# ON STABILITY AND GENERALIZATION OF BILEVEL OPTIMIZATION PROBLEM

## ABSTRACT

(Stochastic) bilevel optimization is a frequently encountered problem in machine learning with a wide range of applications such as meta-learning, hyper-parameter optimization, and reinforcement learning. Most of the existing studies on this problem only focused on analyzing the convergence or improving the convergence rate, while little effort has been devoted to understanding its generalization behaviors. In this paper, we conduct a thorough analysis on the generalization of first-order (gradient-based) methods for the bilevel optimization problem. We first establish a fundamental connection between algorithmic stability and generalization gap in different forms and give a high probability generalization bound which improves the previous best one from $\mathcal{O}(\sqrt{n})$ to $\mathcal{O}(\log n)$, where $n$ is the sample size. We then provide the first stability bounds for the general case where both inner and outer level parameters are subject to continuous update, while existing work allows only the outer level parameter to be updated. Our analysis can be applied in various standard settings such as strongly-convex-strongly-convex (SC-SC), convex-convex (C-C), and nonconvex-nonconvex (NC-NC). Our analysis for the NC-NC setting can also be extended to a particular nonconvex-strongly-convex (NC-SC) setting that is commonly encountered in practice. Finally, we corroborate our theoretical analysis and demonstrate how iterations can affect the generalization gap by experiments on meta-learning and hyper-parameter optimization.

## 1 INTRODUCTION

(Stochastic) bilevel optimization is a widely confronted problem in machine learning with various applications such as meta-learning (Finn et al., 2017; Bertinetto et al., 2018; Rajeswaran et al., 2019), hyper-parameter optimization (Franceschi et al., 2018; Shaban et al., 2019; Baydin et al., 2017; Bergstra et al., 2011; Luketina et al., 2016), reinforcement learning (Hong et al., 2020), and few-shot learning (Koch et al., 2015; Santoro et al., 2016; Vinyals et al., 2016). The basic form of this problem can be defined as follows

$$
\begin{aligned}
\min_{\mathbf{x} \in \mathbb{R}^{d_1}} R(\mathbf{x}) &= F\left(\mathbf{x}, \mathbf{y}^*(\mathbf{x})\right) := \mathbb{E}_\xi\left[f\left(\mathbf{x}, \mathbf{y}^*(\mathbf{x}); \xi\right)\right] \\
\text{s.t. } \mathbf{y}^*(\mathbf{x}) &= \arg\min_{\mathbf{y} \in \mathbb{R}^{d_2}} \left\{G(\mathbf{x}, \mathbf{y}) := \mathbb{E}_\zeta[g(\mathbf{x}, \mathbf{y}; \zeta)]\right\},
\end{aligned}
\tag{1}
$$

where $f : \mathbb{R}^{d_1} \times \mathbb{R}^{d_2} \to \mathbb{R}$ and $g : \mathbb{R}^{d_1} \times \mathbb{R}^{d_2} \to \mathbb{R}$ are two continuously differentiable loss functions with respect to $\mathbf{x}$ and $\mathbf{y}$. Problem (1) has an optimization hierarchy of two levels, where the outer-level objective function $f$ depends on the minimizer of the inner-level objective function $g$.

Due to its importance, the above bilevel optimization problem has received considerable attention in recent years. A natural way to solve problem (1) is to apply alternating stochastic gradient updates with approximating $\nabla_{\mathbf{y}} g(\mathbf{x}, \mathbf{y})$ and $\nabla f(\mathbf{x}, \mathbf{y})$, respectively. Briefly speaking, previous efforts mainly examined two types of methods to perceive an approximate solution that is close to the optimum $y^*(\mathbf{x})$. One is to utilize the single-timescale strategy (Chen et al., 2021; Guo et al., 2021; Khanduri et al., 2021; Hu et al., 2022), where the updates for $\mathbf{y}$ and $\mathbf{x}$ are carried out simultaneously. The other one is to apply the two-timescale strategy (Ghadimi & Wang, 2018; Ji et al., 2021;

| REFERENCE | STABILITY BOUNDS IN VARIOUS SETTINGS | | | |
|---|---|---|---|---|
| | SC-SC | C-C | NC-NC | NC-SC |
| SSGD (THIS WORK) | $\mathcal{O}(1/m_1)$ | $\mathcal{O}(\kappa_1{}^{K/2}/m_1)$ | $\mathcal{O}(K^{\kappa_2}/m_1)$ | $\mathcal{O}(K^{\kappa_3}/m_1)$ |
| TSGD (THIS WORK) | $\mathcal{O}((\kappa_4)^K/m_1)$ | $\mathcal{O}((\kappa_4)^K/m_1)$ | $\mathcal{O}(T^{1-\kappa_5}K^{\kappa_5}/m_1)$ | $\mathcal{O}(T^{1-\kappa_6}K^{\kappa_6}/m_1)$ |

Table 1: Summary of main results. $\kappa_i$: a constant for all $i$ above; $T$: inner iterations; $K$: outer iterations; $m_1$: size of outer dataset. SSGD and TSGD stand for Algorithm 1 and Algorithm 2, the single-timescale and two-timescale methods, via stochastic gradient descent.

Hong et al., 2020; Pedregosa, 2016), where the update of $\mathbf{y}$ is repeated multiple times to achieve a more accurate approximation before conducting the update of $\mathbf{x}$.

While there is a long list of work on bilevel optimization, most of the existing work only focuses on either analyzing its convergence behaviors (Ghadimi & Wang, 2018; Hong et al., 2020; Ji et al., 2021) or improving its convergence rate, based on the convexity and the smoothness properties of $f(\cdot, \cdot)$ and/or $g(\cdot, \cdot)$ (Liu et al., 2020; Li et al., 2020). Contrarily, only little effort is devoted to understanding the generalization behavior of the problem. To the best of our knowledge, there is only one recent work on the generalization analysis for bilevel problems (Bao et al., 2021), which presents the first expected uniform stability bound. However, there are still several undesirable issues in this work: (1) Their result is only for the uniform stability (which could be deduced from argument stability with certain conditions, see Definition 4 for details), leaving the analysis of other stronger definitions of algorithmic stability open; (2) Additionally, the UD algorithm allows the outer level parameters to be updated continuously but needs to reinitialize the inner level parameters before each iteration in the inner loop, which is not commonly used in practice due to their inefficiency (see line 4 in Algorithm 3). (3) The proof of Theorem 2 in their work is unclear to show whether the update of outer level parameters is argument dependent on the inner level parameters, where may exist some gap in the analysis of UD algorithm (see Appendix E for detailed discussions). (4)Their experiments take only hyper-parameter optimization into consideration and neglect other applications in the bilevel optimization instances.

To address all the aforementioned issues, we give in this paper a thorough analysis on the generalization behaviors of first-order (gradient-based) methods for general bilevel optimization problem. We employ the recent advances of algorithmic stability to investigate the generalization behaviors in different settings. Specifically, our main contributions can be summarized as follows:

- Firstly, we establish a fundamental connection between generalization gap and different notations of algorithmic stability (argument stability and uniform stability) for any randomized bilevel optimization algorithms in both expectation and high probability forms. Specifically, we show that the high probability form of the generalization gap bound can be improved from $\mathcal{O}(\sqrt{n})$ to $\mathcal{O}(\log n)$ compared with the result in Bao et al. (2021).

- Next, we present the stability bounds for gradient-based methods with either single-timescale or two-timescale update strategy under different standard settings. To the best of our knowledge, this work provides the first stability bounds for the two-timescale (double loop) algorithms, which allows the accumulation of the sub-sampled gradients in the inner level. In detail, we consider the settings of strongly-convex-strongly-convex (SC-SC), convex-convex (C-C), and nonconvex-nonconvex (NC-NC), and further extend our analysis to a particular nonconvex-strongly-convex (NC-SC) setting that is widely appeared in practice. **Table 1** is the summary of our main results.

- Thirdly, we provide the first generalization bounds for the case where both the outer and inner level parameters are subject to continuous (iterative) changes. Compared to the previous work (Bao et al., 2021), our work does not need the reinitialization step before each iteration in the inner level and hence our algorithm can carry over the last updated inner level parameters, which is more general and practical.

- Finally, we conduct empirical studies to corroborate our theories via meta-learning and hyperparameter optimization, which are two applications of bilevel optimization.

Due to space limitations, all the proofs and additional experiments are included in Appendix.

## 1.1 RELATED WORK

Research at the interface between generalization and the bilevel problem can be roughly classified into two categories. The first one includes all the research on bilevel optimization. In recent decades, extensive studies have been done on this topic, which suggests that bilevel optimization has a wide range of applications in machine learning such as hyper-parameter optimization (Franceschi et al., 2018; Lorraine & Duvenaud, 2018; Okuno et al., 2021), meta learning (Bertinetto et al., 2018; Rajeswaran et al., 2019; Soh et al., 2020) and reinforcement learning (Yang et al., 2018; Tschiatschek et al., 2019). Most of the existing work studies the problem from an optimization perspective. For example, Ghadimi & Wang (2018); Ji et al. (2021) provide the convergence rate analysis based on the nonconvex-strongly-convex assumption for the two functions $f(\cdot, \cdot)$ and $g(\cdot, \cdot)$. (Grazzi et al., 2020) considers the iteration complexity for hypergradient computation. (Liu et al., 2020; Li et al., 2020) present an asymptotic analysis for the convex-strongly-convex setting. Perhaps the most related one to ours from the generalization standpoint (*i.e.,* the expectation of population risk and empirical risk) is Bao et al. (2021), while there may exist some gap in the analysis of UD algorithm. In this work, we employ a novel approach to examine the stability bounds of bilevel optimization problems. Firstly, our work analyzes the generalization behavior by observing how different settings can have an impact on the stability bounds directly. Secondly, our work adopts a stronger version of stability called *argument stability*, which can imply the previously used *uniform stability* if the function is sufficiently smooth. Furthermore, our work does not need to reinitialize the inner-level parameters and allows them to carry over their last updated parameters at each time updating the inner level. This indicates that $\mathbf{y}$ in the inner level is updated iteratively and depends on the current parameter of $\mathbf{x}$, which is more common and efficient in practice.

The second category includes all the work on stability analysis. There is a long list of research on stability and generalization (Bousquet & Elisseeff, 2002; Mukherjee et al., 2006; Shalev-Shwartz et al., 2010). Bousquet & Elisseeff (2002) first introduces the notion of *uniform stability* and establishes the first framework of stability analysis. Hardt et al. (2016) later extends the stability analysis to iterative algorithms based on stochastic gradient methods for the vanilla stochastic optimization. After that, there are subsequent studies on generalization analysis for various problems via algorithmic stability, such as minmax problems (Lei et al., 2021; Farnia & Ozdaglar, 2021; Zhang et al., 2021) and pairwise learning (Yang et al., 2021; Lei et al., 2020; Xue et al., 2021; Huai et al., 2020). However, it is notable that due to the additional stochastic function in the constraint in the bilevel optimization, all the previous techniques and results cannot be applied to our problem. Although the generalization analysis of minmax optimization is somewhat similar to ours, it involves only one objective function $f$ and a single level in algorithms for typical minmax optimization problems, while in the bilevel optimization algorithms there is an inner level and an outer level, which is considerably more challenging.

## 2 PRELIMINARIES

### 2.1 DEFINITIONS AND ASSUMPTIONS

In the following, we give some necessary definitions and assumptions that are widely used in bilevel optimization (Ghadimi & Wang, 2018; Ji et al., 2021; Khanduri et al., 2021) and generalization analysis (Hardt et al., 2016; Lei et al., 2021).

**Definition 1** (Joint Lipschitz Continuity)**.** A function $f(\mathbf{x}, \mathbf{y})$ is jointly $L$-Lipschitz over $\mathbb{R}^{d_1} \times \mathbb{R}^{d_2}$, if for all $\mathbf{x} \in \mathbb{R}^{d_1}, \mathbf{y} \in \mathbb{R}^{d_2}$, the following holds, $|f(\mathbf{x}, \mathbf{y}) - f(\mathbf{x}', \mathbf{y}')| \leq L\sqrt{\|\mathbf{x} - \mathbf{x}'\|_2^2 + \|\mathbf{y} - \mathbf{y}'\|_2^2}$.

**Definition 2** (Smoothness)**.** A function $f$ is $l$-smooth over a set $S$ if for all $u, w \in S$ the following is true, $\|\nabla f(u) - \nabla f(w)\| \leq l\|u - w\|$.

**Definition 3** (Strong Convexity)**.** A function $f$ is $\mu$-strongly-convex over a set $S$, if for all $u, w \in S$, the following holds, $f(u) + \langle \nabla f(u), w - u \rangle + \frac{\mu}{2}\|w - u\|^2 \leq f(w)$.

**Assumption 1** (Inner-level Function Assumption)**.** We assume the inner stochastic function $g(\mathbf{x}, \mathbf{y})$ in (1) satisfies the following:
(*i*) $g(\mathbf{x}, \mathbf{y})$ is jointly $L_g$-Lipschitz for any $\mathbf{x} \in \mathbb{R}^{d_1}$ and $\mathbf{y} \in \mathbb{R}^{d_2}$.
(*ii*) $g(\mathbf{x}, \mathbf{y})$ is continuously differentiable and $l_g$-smooth for any $(\mathbf{x}, \mathbf{y}) \in \mathbb{R}^{d_1} \times \mathbb{R}^{d_2}$.

**Assumption 2** (Outer-level Function Assumption)**.** We assume the outer stochastic function $f(\mathbf{x}, \mathbf{y})$ in (1) satisfies the following:

*(iii)* $f(\mathbf{x}, \mathbf{y})$ is jointly $L_f$-Lipschitz for any $\mathbf{x} \in \mathbb{R}^{d_1}$ and $\mathbf{y} \in \mathbb{R}^{d_2}$.

*(iv)* $f(\mathbf{x}, \mathbf{y})$ is continuously differentiable and $l_f$-smooth for any $(\mathbf{x}, \mathbf{y}) \in \mathbb{R}^{d_1} \times \mathbb{R}^{d_2}$.

## 2.2 PROBLEM FORMULATION

Given two distributions $\mathbb{D}_1$ and $\mathbb{D}_2$, in the (stochastic) optimization problem we aim to find the minimizer of Problem (1). However, since the distributions are often unknown, in practice we only have two finite-size datasets $D_{m_1} = \{\xi_i \mid i = 1, ..., m_1\} \sim \mathbb{D}_1^{m_1}$ and $D_{m_2} = \{\zeta_i \mid i = 1, ..., m_2\} \sim \mathbb{D}_2^{m_2}$, where each $\xi_i$ and $\zeta_i$ are i.i.d. sampled from $\mathbb{D}_1$ and $\mathbb{D}_2$, respectively. Based on these datasets, we will design some (randomized) algorithm $\mathbf{A}$ with output $\mathbf{A}(D_{m_1}, D_{m_2}) = (\mathbf{x}, \mathbf{y}) \in \mathbb{R}^{d_1} \times \mathbb{R}^{d_2}$. Our goal is to investigate the generalization behavior of such output. Note that although there are two stochastic functions in the bilevel optimization problem, we only care about the generalization of the outer-level one since it is the one that we prefer to minimize.

Below we define the generalization gap to measure the generalization behavior. Given distribution $\mathbb{D}_1$ and a finite data $D_{m_1} \sim \mathbb{D}_1^{m_1}$, the population risk function $R(\mathbf{x}, \mathbf{y}, \mathbb{D}_1)$ of $\mathbf{x}$, $\mathbf{y}$ on $\mathbb{D}_1$ is defined as $R(\mathbf{x}, \mathbf{y}, \mathbb{D}_1) := \mathbb{E}_{\xi \sim \mathbb{D}_1} [f(\mathbf{x}, \mathbf{y}(\mathbf{x}); \xi)]$, and its empirical risk function on $D_{m_1}$ is $R_s(\mathbf{x}, \mathbf{y}, D_{m_1}) = \frac{1}{m_1} \sum_{i=1}^{m_1} [f(\mathbf{x}, \mathbf{y}(\mathbf{x}); \xi_i)]$. Moreover, for a fixed hyperparameter $\mathbf{x} \in \mathbb{R}^{d_1}$ and $\mathbf{y}(\mathbf{x}) \in \mathbb{R}^{d_2}$ ( note that $\mathbf{y}(\mathbf{x})$ might be dependent on $\mathbf{x}$), we define the difference between the population risk and the empirical risk over $(\mathbf{x}, \mathbf{y}(\mathbf{x}))$ as the *bilevel generalization gap of* $(\mathbf{x}, \mathbf{y}(\mathbf{x}))$: $\mathbb{E}_s[R(\mathbf{x}, \mathbf{y}) - R_s(\mathbf{x}, \mathbf{y})]$, where $\mathbb{E}_s$ denotes the expectation of $D_{m_1} \sim \mathbb{D}_1^{m_1}$. When there is no ambiguity, we simplify thereafter the notations as follows: $R(\mathbf{x}, \mathbf{y}, \mathbb{D}_1) = R(\mathbf{x}, \mathbf{y})$ and $R_s(\mathbf{x}, \mathbf{y}, D_{m_1}) = R_s(\mathbf{x}, \mathbf{y})$. Our goal is thus to analyze the *bilevel generalization gap of the output of algorithm* $\mathbf{A}(D_{m_1}, D_{m_2})$ based on $D_{m_1}$ and $D_{m_2}$. Since the generalized error depends on the algorithm itself, in the following we will introduce the algorithms to be considered in this paper.

Most of the existing algorithms adopt the following idea: first approximate $\mathbf{y}^*$ on $D_{m_2}$ for a given parameter $\mathbf{x}$ in the inner level and then seek the hyperparameter $\mathbf{x}^*(D_{m_1}, D_{m_2})$ with corresponding hypothesis $\mathbf{y}^*(\mathbf{x}^*(D_{m_1}, D_{m_2}), D_{m_2})$ by the below estimation:

$$\hat{\mathbf{x}}(D_{m_1}, D_{m_2}) \approx \arg\min_{\mathbf{x}} R_s(\mathbf{x}, \hat{\mathbf{y}}(\mathbf{x}, D_{m_2}), D_{m_1}),$$

$$\text{where} \quad \hat{\mathbf{y}}(\mathbf{x}, D_{m_2}) \approx \arg\min_{\mathbf{y}} G_s(\mathbf{x}, \mathbf{y}, D_{m_2}), \tag{2}$$

where $G_s(\mathbf{x}, \mathbf{y}, D_{m_2})$ is the empirical risk of $G(\mathbf{x}, \mathbf{y})$ over $D_{m_2}$, *i.e.,* $G(\mathbf{x}, \mathbf{y}, D_{m_2}) = \frac{1}{m_2} \sum_{i=1}^{m_2} g(\mathbf{x}, \mathbf{y}(\mathbf{x}); \zeta_i)$. Most of the current gradient-based (first-order) algorithms for approximating (2) can be categorized into two classes: single-timescale methods and two-timescale methods. The single-timescale method performs the updates for $\mathbf{y}$ and $\mathbf{x}$ simultaneously via stochastic gradient descent (SGD), while the two-timescale method updates $\mathbf{y}$ multiple times before updating $\mathbf{x}$ (via stochastic gradient descent). As there are numerous approaches for both classes (see Related Work section for details), in this paper we will analyze the generalization behaviors for the most classical and standard one in each class, *i.e.,* single-timescale SGD (SSGD; Algorithm 1) and two-timescale SGD (TSGD; Algorithm 2). There is a long list of work (Chen et al., 2021), (Ghadimi & Wang, 2018; Ji et al., 2021) based on either SSGD or TSGD.

## 3 GENERALIZATION AND STABILITY FOR BILEVEL OPTIMIZATION

Algorithmic stability is one of the classical approaches to analyzing the generalization bound for algorithms. Roughly speaking, the algorithmic stability of (randomized) algorithm $\mathbf{A}$ measures how the output of algorithm $\mathbf{A}$ changes if we change one data sample in the input dataset. While there are various notions of stability, most of the existing work on analyzing the stability of stochastic optimization, pairwise learning and minimax optimization focuses on the uniform-stability (Bousquet & Elisseeff, 2002) and the argument-stability (Liu et al., 2017; Lei & Ying, 2020). Thus, we also adopt these two notions of stability for the bilevel optimization problem. Briefly speaking, uniform-stability focuses on the resulting change in population risk function, while the argument-stability considers the resulting change in arguments, *i.e.,* the output of the algorithm.

**Definition 4** (Algorithmic Stability)**.** Let $\mathbf{A}: \mathbb{D}_1^{m_1} \times \mathbb{D}_2^{m_2} \mapsto \mathbb{R}^{d_1} \times \mathbb{R}^{d_2}$ be a randomized algorithm.

| **Algorithm 1** Single-timescale SGD (SSGD) | **Algorithm 2** Two-timescale SGD (TSGD) |
|---|---|
| 1: **Input:** number of iterations $K$, step sizes $\alpha_x$, $\alpha_y$, initialization $\mathbf{x}_0, \mathbf{y}_0$, Datasets $D_{m_1}$ and $D_{m_2}$ | 1: **Input:** number of iterations $K$, step sizes $\alpha_x$, $\alpha_y$, initialization $\mathbf{x}_0, \mathbf{y}_0$ |
| 2: **Output:** $\mathbf{x}_K, \mathbf{y}_K$ | 2: **Output:** $\mathbf{x}_K, \mathbf{y}_K$ |
| 3: **for** $k = 0$ **to** $K - 1$ **do** | 3: **for** $k = 0$ **to** $K - 1$ **do** |
| 4: $\quad$ Uniformly sample $i \in [m_2], j \in [m_1]$ | 4: $\quad \mathbf{y}_k^0 \leftarrow \mathbf{y}_{k-1}^T$ |
| 5: $\quad \mathbf{y}_{k+1} = \mathbf{y}_k - \alpha_y \nabla_\mathbf{y} g(\mathbf{x}_k, \mathbf{y}_k(\mathbf{x}_k); \zeta_i)$ | 5: $\quad$ **for** $t = 0$ **to** $T - 1$ **do** |
| 6: $\quad \mathbf{x}_{k+1} = \mathbf{x}_k - \alpha_x \nabla f(\mathbf{x}_k, \mathbf{y}_k(\mathbf{x}_k); \xi_j)$ | 6: $\quad\quad$ Uniformly sample $i \in [m_2]$ |
| 7: **end for** | 7: $\quad\quad \mathbf{y}_k^{t+1} = \mathbf{y}_k^t - \alpha_y \nabla_\mathbf{y} g(\mathbf{x}_k, \mathbf{y}_k^t(\mathbf{x}_k); \zeta_i)$ |
| 8: **return** $\mathbf{x}_K$ and $\mathbf{y}_K$ | 8: $\quad$ **end for** |
| | 9: $\quad$ Uniformly sample $j \in [m_1]$ |
| | 10: $\quad \mathbf{x}_{k+1} = \mathbf{x}_k - \alpha_x \nabla f(\mathbf{x}_k, \mathbf{y}_k^T(\mathbf{x}_k); \xi_j)$ |
| | 11: **end for** |
| | 12: **return** $\mathbf{x}_K, \mathbf{y}_K^T$ |

(a) $\mathbf{A}$ is $\beta$-uniformly-stable if for all datasets $D_{m_1}, D'_{m_1} \sim \mathbb{D}_1^{m_1}$ and $D_{m_2} \sim \mathbb{D}_2^{m_2}$ such that $D_{m_1}$ and $D'_{m_1}$ differ in at most one sample, we have the following for any $\xi \sim \mathbb{D}_1$:

$$\mathbb{E}_\mathbf{A}[|f(\mathbf{A}(D_{m_1}, D_{m_2}), \xi) - f(\mathbf{A}(D'_{m_1}, D_{m_2}), \xi)|] \leq \beta.$$

$\mathbf{A}$ is $\beta$-uniformly-stable with probability at least $1 - \delta$ if we have the following for any $\xi \sim \mathbb{D}_1$ with probability at least $1 - \delta$:

$$\left| f(\mathbf{A}(D_{m_1}, D_{m_2}), \xi) - f(\mathbf{A}(D'_{m_1}, D_{m_2}), \xi) \right| \leq \beta.$$

(b) $\mathbf{A}$ is $\beta$-argument-stable in expectation if for all datasets $D_{m_1}, D'_{m_1} \sim \mathbb{D}_1^{m_1}$ and $D_{m_2} \sim \mathbb{D}_2^{m_2}$ such that $D_{m_1}$ and $D'_{m_1}$ differ in at most one sample, we have:

$$\mathbb{E}_\mathbf{A}[\|\mathbf{A}(D_{m_1}, D_{m_2}) - \mathbf{A}(D'_{m_1}, D_{m_2})\|_2] \leq \beta.$$

Note that the definition of uniform stability in expectation is the same as the definition in (Bao et al., 2021). Thus, our other definitions can be considered as extensions of the previous stability for bilevel optimization. In the following, we present Theorem 1 as our first result, which shows a crucial relationship between generalization gap and algorithmic stability for an algorithm $\mathbf{A}$.

**Theorem 1.** *Let* $\mathbf{A} : \xi^{m_1} \times \zeta^{m_2} \mapsto \mathbb{R}^{d_1} \times \mathbb{R}^{d_2}$ *be a randomized BO algorithm.*

(a) *If* $\mathbf{A}$ *is* $\beta$-*uniform-stable in expectation, then the following holds for* $D_{m_1} \sim \mathbb{D}_1^{m_1}, D_{m_2} \sim \mathbb{D}_2^{m_2}$:

$$\mathbb{E}_{\mathbf{A}, D_{m_1}}[R(\mathbf{A}(D_{m_1}, D_{m_2})) - R_s(\mathbf{A}(D_{m_1}, D_{m_2}))] \leq \beta.$$

(b) *If* $\mathbf{A}$ *is* $\beta$-*argument-stable in expectation and Assumption 2 holds, then the following holds for* $D_{m_1} \sim \mathbb{D}_1^{m_1}, D_{m_2} \sim \mathbb{D}_2^{m_2}$:

$$\mathbb{E}_{\mathbf{A}, D_{m_1}}[R(\mathbf{A}(D_{m_1}, D_{m_2})) - R_s(\mathbf{A}(D_{m_1}, D_{m_2}))] \leq L_f \beta.$$

(c) *Assume that* $|f(\mathbf{x}, \mathbf{y}; \xi)| \leq M$ *for some* $M \geq 0$. *If* $\mathbf{A}$ *is* $\beta$-*uniform-stable almost surely, then for* $D_{m_1} \sim \mathbb{D}_1^{m_1}, D_{m_2} \sim \mathbb{D}_2^{m_2}$, *the following holds with probability* $1 - \delta$:

$$|R(\mathbf{A}(D_{m_1}, D_{m_2})) - R_s(\mathbf{A}(D_{m_1}, D_{m_2}))|$$
$$\leq 2\beta + e\left(\frac{4M}{\sqrt{m_1}}\sqrt{\log\frac{e}{\delta}} + 12\sqrt{2}\beta\lceil\log_2 m_1\rceil\sqrt{\log\frac{e}{\delta}}\right)$$

*where e is the base of the natural logarithms.*

**Remark 1.** The above theorem suggests that the generalization gap can be controlled by several notions of algorithmic stability. Part (a) and Part (b) show that the expectation of generalization gap can be bounded by uniform stability and argument stability with the Lipschitz constant, respectively; Part (c) indicates that the generalization gap for the algorithm is no more than $\mathcal{O}(\beta \log(m_1) +$

$1/\sqrt{m_1}$) with probability $1 - \delta$. Compared with the existing work (Bao et al., 2021), Theorem 1 considers argument stability additionally, which is a stronger notion of stability than uniform stability (since uniform stability can be deduced from argument stability with the condition that the function is sufficiently smooth). Moreover, we use the McDiarmid's inequality and the equivalence of tails and moments for the random variable with a mixture of sub-gaussian and sub-exponential tails (Lemma 1 in Bousquet et al. (2020)), which provide a significantly improved high probability bound in Part (c) (*i.e.,* improving from $\mathcal{O}(\beta\sqrt{m_1})$ in Bao et al. (2021) to $\mathcal{O}(\beta \log m_1)$).

## 4 STABILITY ANALYSIS FOR BILEVEL OPTIMIZATION ALGORITHMS

Motivated by Theorem 1, we can see that to analyze the generalization behaviors for any algorithm, it is sufficient to analyze its stability. As mentioned in the previous Section 2.2, we will consider the stability of SSGD and TSGD. For simplicity we let SC-SC denote the case where $f$ and $g$ both are strongly convex functions. C-C, NC-NC, and NC-SC are also denoted in a similar manner with "C" representing convex function and "NC" representing nonconvex function.

### 4.1 STABILITY BOUNDS FOR SINGLE-TIMESCALE SGD

As we can see from Algorithm 1, SSGD updates $\mathbf{y}$ and $\mathbf{x}$ simultaneously. In the following we develop stability bounds for this algorithm in different settings.

**Theorem 2.** *Suppose that Assumptions 1 and 2 hold and Algorithm* $\mathbf{A}$ *is SSGD with* $K$ *iterations:*

(a) *Assume that Problem (1) is SC-SC with strongly convexity parameters* $\mu_f$ *and* $\mu_g$. *Let* $\alpha_x = \alpha_y$ *(see Lemma 9 for details) be the step sizes. Denote* $l = \max\{l_f, l_g\}$. *Then,* $\mathbf{A}$ *is* $\beta$-*argument-stable in expectation, where*

$$\beta \leq \mathcal{O}\left( \left( L_f^2 + L_g^2 \right)^{\frac{1}{2}} \left( m_1 \left( \mu_f + \mu_g - (\alpha_x l)^2 / 2 + 0.25 \right) \right)^{-1} \right).$$

(b) *Assume that Problem (1) is C-C. Let* $\alpha_x$, $\alpha_y$ *be the step sizes. Then,* $\mathbf{A}$ *is* $\beta$-*argument-stable in expectation, where*

$$\beta \leq \mathcal{O}\left( m_1^{-1} \sqrt{\left( \alpha_x L_f \right)^2 + \left( \alpha_y L_g \right)^2} \left( 2 + 2 \max\left\{ \left( \alpha_x l_f \right)^2, \left( \alpha_y l_g \right)^2 \right\} \right)^{K/2} \right).$$

(c) *Assume that Problem (1) is NC-NC. Let the step sizes satisfy* $\max\left\{ \alpha_x, \alpha_y \right\} \leq c/k$ *for some constant* $c \geq 0$ *and* $l = \max\left\{ l_f, l_g \right\}$. *Then,* $\mathbf{A}$ *is* $\beta$-*argument-stable in expectation, where*

$$\beta \leq \mathcal{O}\left( (m_1 cl)^{-1} \left( 2cL_f \sqrt{l_f^2 + l_g^2} \right)^{\frac{1}{cl+1}} \cdot K^{\frac{cl}{cl+1}} \right),$$

*where* $l_f$, $l_g$ *and* $L_f$, $L_g$ *are smoothness constants and Lipschitz constants for* $f$, $g$, *respectively.*

**Remark 2.** Note that the above stability bounds are independent of the specific form of the objective function $f(\cdot, \cdot)$ and the exact form of the sample distribution $\mathbb{D}_1$, which are more reliant on the properties of the loss functions and sample size $m_1$, and the stability bounds in the C-C and NC-NC cases are related to the number of iterations additionally. Specifically, Part(a) establishes a stability bound of $\mathcal{O}(1/m_1)$ in the SC-SC setting and Part(b) considers a C-C case with a stability bound $\mathcal{O}(\kappa_1^{K/2}/m_1)$ related to the number of iterations and the data size, where $\kappa_1$ is a constant. The NC-NC case is discussed in Part(c) which provides a stability bound of $\mathcal{O}(K^{\frac{cl}{cl+1}}/m_1)$, where $c$ is a constant to control the step size and $l$ is the larger smoothness number of $l_f$ and $l_g$. The conclusions here match the existing results in minmax problems (Lei et al., 2021; Farnia & Ozdaglar, 2021).

### 4.2 STABILITY BOUNDS FOR TWO-TIMESCALE SGD

Compared with the above SSGD, Two-timescale SGD (TSGD; Algorithm 2) always achieves more accurate approximate solutions by updating $\mathbf{y}$ multiple times before updating $\mathbf{x}$. In this section, we extend our analysis from SSGD to TSGD. Particularly, compared with the results in Bao et al. (2021), we provide stability bounds in Theorem 3 for the case where the inner level parameter

(**y**) is updated iteratively (*i.e.*, consistency). We further explore in Theorem 4 a particular NC-SC setting, which is commonly appeared in bilevel optimization applications such as meta learning and hyperparameter optimization.

**Theorem 3.** *Suppose that Assumptions 1 and 2 hold and $|g(\cdot, \cdot)| \leq 1$. Let $\mathbf{A}$ be the TSGD algorithm with $K$ outer-iterations and $T$ inner-iterations. Then we have*

(a) *Assume that Problem (1) is SC-SC. Let $l = \max\{l_f, \frac{1+(\alpha_y l_g)^2}{(1-\alpha_y l_g)\alpha_y}\}$ and $\alpha = \alpha_x = \alpha_y \leq \min\{1/l_g, 1/(\mu_f + \mu_g)\}$ be the step sizes. Then, $\mathbf{A}$ is $\beta$-argument-stable in expectation, where*

$$\beta \leq \mathcal{O}\left(m_1^{-1}\sqrt{L_f^2\alpha_x^2 + \left(\frac{2T}{\alpha_y(2-\alpha_y l_g)}\right)^2}(1+\alpha l)^K\right).$$

(b) *Assume that Problem (1) is C-C. Let $\alpha l = \max\{\alpha_x l_f, \frac{1+(\alpha_y l_g)^2}{1-\alpha_y l_g}\}$ and $\alpha_x, \alpha_y \leq \frac{1}{l_g}$ be the step sizes. Then, $\mathbf{A}$ is $\beta$-argument-stable in expectation, where*

$$\beta \leq \mathcal{O}\left(m_1^{-1}\sqrt{L_f^2\alpha_x^2 + \left(\frac{2T}{\alpha_y(2-\alpha_y l_g)}\right)^2}(1+\alpha l)^K\right).$$

(c) *Assume that Problem (1) is NC-NC. Let the step sizes satisfy $\max\{\alpha_x, \alpha_y\} \leq c/k$ for some constant $c \geq 0$ and $l = \max\{l_f, l_g\}$. Then, $\mathbf{A}$ is $\beta$-argument-stable in expectation, where*

$$\beta \leq \mathcal{O}\left((m_1 Tcl)^{-1}\left(2cL_f\sqrt{l_f^2 + T^2 l_g^2}\right)^{\frac{1}{Tcl+1}} \cdot K^{\frac{Tcl}{Tcl+1}}\right).$$

**Remark 3.** Compared with the previous results for SSGD, the stability bounds of TSGD depend on the number of iterations in the outer level loop, the number of iterations in the inner level loop, and the data size in the outer level loop. If the step sizes are sufficiently small, we can see that the bounds in Theorem 3 are asymptotically the same as the bounds of SSGD in Theorem 2. Thus, Theorem 3 can be considered as a generalization of the previous one. The dependence on $T$ also reveals our novelty compared with the existing work of stability analysis for other problems, such as simple SGD and minmax problems. To the best of our knowledge, this work provides the first stability bounds for the two-timescale (double loop) algorithms, which allows the accumulation of the sub-sampled gradients in the inner level.

**Remark 4.** Comparing our results with the ones in (Bao et al., 2021), we have the following observations. 1) They only established the uniform stability bound for the Unrolled Differentiation algorithm 3, where the algorithm is reinitialized at each time entering the inner level loop, indicating that it takes into account the changes to only one parameter in the outer level loop, while our algorithm considers the update for both parameters. 2) Its proof needs to assume that the update of $y$ in the inner level after the reinitialization will not be affected by the value specified for **x**. However, this assumption is quite uncommon and is probably the reason that they do not need to make any assumption on the inner level objective function (see Appendix E in details). In contrast, our work allows the inner level parameters to be updated consistently (*i.e.*, carrying over the value in the last update), *instead of being reinitialized at each time entering the inner level loop*. Specifically, we allow $y_k^T$ to be employed at the beginning of the $(k+1)$-th outer level iteration, rather than $y_0$. This enables us to obtain different stability bounds for different inner level objective functions from a novel perspective.

In the following, we extend our analysis to a particular NC-SC setting that is frequently encountered in real-world applications and optimization analysis.

**Theorem 4.** *Suppose that Assumptions 1 and 2 hold, $0 \leq f(\cdot, \cdot) \leq 1$ and Problem (1) is NC-SC. Let $\mathbf{A}$ be the TSGD Algorithm with $K$ outer-iterations and $T$ inner-iterations with $\max\{\alpha_x, \alpha_y\} \leq c/k$ for constant $c \geq 0$. Denote $l = \max\{l_f, l_g\}$. Then, $\mathbf{A}$ is $\beta$-uniform-stable in expectation, where*

$$\beta \leq \mathcal{O}\left(\frac{\left(2cL_f\sqrt{l_f^2 + l_g^2 T^2}\right)^{\frac{1}{c(Tl+l-\mu_g)+1}} \cdot K^{\frac{c(Tl+l-\mu_g)}{c(Tl+l-\mu_g)+1}}(Tl+l-\mu_g+2/c)}{m_1(Tl+l-\mu_g)}\right).$$

**Remark 5.** Compared with our previous analysis, we now sketch the technique differences in our analysis. We consider the bound of the term $(\delta_{\mathbf{x},k}, \delta_{\mathbf{y},k})^T = (\|\mathbf{x}_k - \mathbf{x}'_k\|, \|\mathbf{y}_k - \mathbf{y}'_k\|)^T$, while we employ $\delta_k = \sqrt{\|\mathbf{x}_k - \mathbf{x}'_k\|_2^2 + \|\mathbf{y}_k - \mathbf{y_k}'\|_2^2}$ in the previous analysis, where $(\mathbf{x}_k, \mathbf{y}_k)$, $(\mathbf{x}'_k, \mathbf{y}'_k)$ are the outputs of TSGD after $k$ iterations for $D_{m_1}$ and $D'_{m_1}$ respectively with $D_{m_1}$ and $D'_{m_1}$ differing in one sample. In the NC-SC setting, we show that $(\delta_{\mathbf{x},k+1}, \delta_{\mathbf{y},k+1})^T \leq ((1 + \alpha_x l)\delta_{\mathbf{x},k}, (1 + \alpha_x Tl)\delta_{\mathbf{y},k})^T$ ($\leq$ means the entry-wise inequality), which means our term can be controlled. Then, we take the expectation of it to derive our uniform stability bound. To achieve the generalization gap over continuously changing parameters, it is imperative to take into account the growth of $(\delta_{\mathbf{x},k}, \delta_{\mathbf{y},k})$ instead of $\delta_{\mathbf{x},k}$ in (Bao et al., 2021). Appendix C.3 provides more details.

Thus, based on our previous results, we now provide the first generalization bounds in the NC-NC setting for both SSGD and TSGD.

**Corollary 5.** *Assume that the problem is NC-NC, $|f(\cdot, \cdot; \xi)| \leq 1$ for all $\xi$, and Assumptions 1 and 2 hold. Denote $l = \max\{l_f, l_g\}$ with $\max\{\alpha_x, \alpha_y\} \leq c/k$ for constant $c \geq 0$. Then, the generalization gap of SSGD 1 with $K$ iterations is bounded by $\mathcal{O}(K^{\frac{cl}{cl+1}}/m_1)$.*

**Corollary 6.** *Assume that the problem is NC-NC, $|f(\cdot, \cdot; \xi)| \leq 1$ for all $\xi$, and Assumptions 1 and 2 hold. Let $l = \max\{l_f, l_g\}$ with $\max\{\alpha_x, \alpha_y\} \leq c/k$. Then, the generalization gap of TSGD 2 with $K$ outer iterations and $T$ inner iterations is bounded by $\mathcal{O}(T^{\frac{1}{Tcl+1}} K^{1 - \frac{1}{Tcl+1}}/(m_1))$.*

**Remark 6.** By Theorem [1, 2, 3], we can derive the above corollaries on generalization gap from stability bounds. Corollary 5 and Corollary 6 show that extremely high number of iterations ($K$ for SSGD and $K$, $T$ for TSGD) will drastically reduce the stability of these algorithms and increase the generalization gap, which will make these algorithms increase the risk of overfitting. We will also verify it in the following experiments.

# 5 EXPERIMENTS

In this section, we empirically validate our previous theoretical results on real world datasets. Two experiments, including meta-learning and hyperparameter optimization, are conducted via Algorithm 2 TSGD (note that when $T = 1$, TSGD is just SSGD). Due to the space limitation, we just present the meta learning experiment here, leaving the hyperparameter optimization experiment and other details in the Appendix D.

## 5.1 META LEARNING

Consider the few-shot meta-learning problem with $M$ tasks $\{\mathcal{T}_i, i = 1, ..., M\}$ sampled from distribution $P_{\mathcal{T}}$. We aim to learn a model that can rapidly adapt to different tasks. Firstly, the embedding model $\phi$ is shared by all tasks to learn embedded features. Secondly, the task-specific parameter $w_i$ is to adapt the shared embedding to its own sub-problem. Thus, the overall problem of meta-learning can be formulated as follow:

$$\min_{\phi} \mathcal{L}_{\mathcal{D}}(\phi, \bar{w}^*) = \mathbb{E}_{\xi \in \mathcal{D}_i^{\text{te}}, \mathcal{T}_i} [\mathcal{L}(\phi, w_i^*; \xi)], \tag{3a}$$

$$\text{s.t. } \bar{w}^* = \arg\min_{\bar{w}} \left[ \mathcal{L}_{\mathcal{D}^{\text{tr}}}(\phi, \bar{w}) = \mathbb{E}_{\mathcal{T}_i} \left[ \mathcal{L}_{\mathcal{D}_i^{\text{tr}}}(\phi, w_i) \right] \right]. \tag{3b}$$

where $\mathcal{D}_i^{\text{tr}}$ and $\mathcal{D}_i^{\text{te}}$ are the training and testing datasets for task $\mathcal{T}_i$. Each $w_i$ is computed from one or more gradient descent updates from $\bar{w}$ on the corresponding task (rapid adaptation), *i.e.,* $w_i = \bar{w} - \alpha \nabla_{\bar{w}} \mathcal{L}_{\mathcal{D}_{tr}}(\phi, w_i)$. In the inner level, the base learner optimizes the series of $w_i$ for each tasks (Equation 3b). In the outer level, the meta-learner optimizes the embedding model $\phi$ using the minimizers $w_i^*$ learned from the inner level and computes the loss from the testing dataset (Equation 3a).

**Settings and Implementation** We evaluate the behavior of the 5-way-1-shot task on the Omnilot dataset (Lake et al., 2015), *i.e.,* it aims to classify 5 unseen classes from only 1 labeled sample. It contains 1623 different handwritten characters from 50 different alphabets. The image is in greyscale with a size $28 \times 28$. We follow similar settings in Ji et al. (2021). A five-layer fully-connected network is constructed, where the task-specific parameter $w_i$ corresponds to the last layer of the

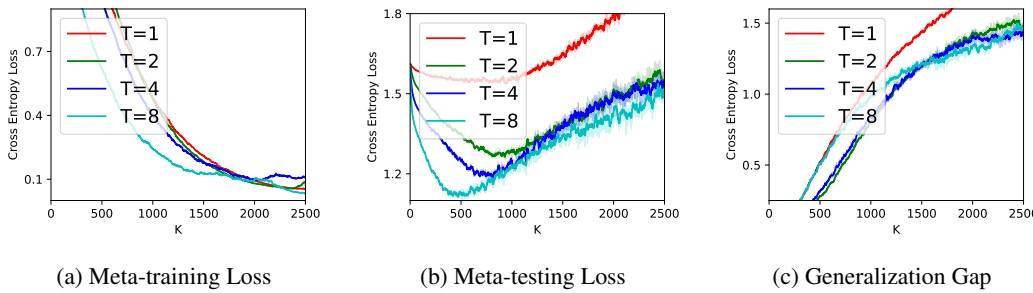

Figure 1: Results of meta learning with various values of $T$ and $K$

network and the shared embedding model $\phi$ corresponds to all preceding layers. Thus, we train the two sets of layers separately in the outer and inner level of optimization. We build our model and establish our training using the software library learn2learn (Arnold et al., 2020). We follow the official train-validation-test partition and train $\phi$, $w_i$ using the training set. The size of each layer in the network is $784 \rightarrow 256 \rightarrow 128 \rightarrow 64 \rightarrow 64 \rightarrow 5$. We set the number of tasks for training and testing set to 2000 and the batch size of tasks to 32. The learning rate of $\phi$ and $w_i$ are 0.002 and 0.01, respectively. Results are evaluated based on the average of 5 trial runs with different random seeds.

**Results Evaluation** Figure 1 presents the learning curves on training set, testing set and the generalization gap with different values of inner iterations $T$ and outer iterations $K$. Generalization gap is estimated by the difference between training and testing loss. On one hand, it can be seen that the model easily overfits on the testing set as $K$ increases drastically (Figure 1b) and the effect of $T$ is very limited. On the other hand, with an appropriate value of $K$, smaller $T$ (*i.e*) will result in underfitting on the testing loss ($T = 1$ in the Figure 1c causes highest generalization gap due to the underfitting training process). The trend of generalization gap in terms of $K$ and $T$ indicates that large values of iteration numbers will increase the risk of overfitting, which matches with our analysis in Theorem 4 that the stability of TSGD 2 will decrease drastically.

## 6 CONCLUSION

We give a thorough analysis on the generalization of first-order (gradient-based) methods for the bilevel optimization framework. In particular, we establish a quantitative connection between generalization and algorithmic stability and provide the first generalization bounds of the continuous updates for inner parameters and outer parameters in multiple settings. Our experiments suggest that inappropriate iterations will cause underfitting and overfitting easily. The tendency of generalization gap also validates our theoretical results.

From the discussion in previous sections, we only discussed the first-order method, while there exist a number of estimating second-order and momentum-based approaches to solve the bilevel optimization problem. Dealing with the approximation of hypergradient in generalization analysis is another direction for future work.

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

## A    COMPARISON BETWEEN UD AND TSGD

| **Algorithm 3** Unrolled differentiation (UD) | **Algorithm 4** Two-timescale SGD (TSGD) |
|---|---|
| 1: **Input:** number of iterations $K$, step sizes $\alpha_x$, $\alpha_y$, initialization $\mathbf{x}_0, \mathbf{y}_0$ | **Input:** number of iterations $K$, step sizes $\alpha_x$, $\alpha_y$, initialization $\mathbf{x}_0, \mathbf{y}_0$, Datasets: $D_{m_1}, D_{m_2}$ |
| 2: **Output:** $\mathbf{x}_K, \mathbf{y}_K$ | |
| 3: **for** $k = 0$ **to** $K - 1$ **do** | **Output:** $\mathbf{x}_K, \mathbf{y}_K$ |
| 4:    $\mathbf{y}_k^0 \leftarrow \mathbf{y}_0$ | **for** $k = 0$ **to** $K - 1$ **do** |
| 5:    **for** $t = 0$ **to** $T - 1$ **do** | $\quad \mathbf{y}_k^0 \leftarrow \mathbf{y}_{k-1}^T$ |
| 6:       $\mathbf{y}_k^{t+1} = \mathbf{y}_k^t - \alpha_y \nabla_\mathbf{y} g(\mathbf{x}_k, \mathbf{y}_k^t(\mathbf{x}_k); D_{m_2})$ | $\quad$ **for** $t = 0$ **to** $T - 1$ **do** |
| 7:    **end for** | $\qquad \mathbf{y}_k^{t+1} = \mathbf{y}_k^t - \alpha_y \nabla_\mathbf{y} g(\mathbf{x}_k, \mathbf{y}_k^t(\mathbf{x}_k); D_{m_2})$ |
| 8:    $\mathbf{x}_{k+1} = \mathbf{x}_k - \alpha_x \nabla f(\mathbf{x}_k, \mathbf{y}_k^T(\mathbf{x}_k); D_{m_1})$ | $\quad$ **end for** |
| 9: **end for** | $\quad \mathbf{x}_{k+1} = \mathbf{x}_k - \alpha_x \nabla f(\mathbf{x}_k, \mathbf{y}_k^T(\mathbf{x}_k); D_{m_1})$ |
| 10: **return** $\mathbf{x}_K, \mathbf{y}_K^T$ | **end for** |
| | **return** $\mathbf{x}_K, \mathbf{y}_K^T$ |

## B    PROOF OF PRELIMINARIES

### B.1    THE PROOF OF THEOREM 1

*Proof of Part (a).* Since $\xi$ and $\xi_i$ are drawn from the same distribution, we know

$$\mathbb{E}_\mathbf{A}[R(\mathbf{A}(D_{m_1}, D_{m_2}), \mathbb{D}_1) - R_s(\mathbf{A}(D_{m_1}, D_{m_2}), D_{m_1})]$$
$$= \mathbb{E}_{\mathbf{A}, \xi_i \in D_{m_1}, \xi \sim \mathbb{D}_1}[f(\mathbf{A}(D_{m_1}, D_{m_2}), \xi) - f(\mathbf{A}(D_{m_1}, D_{m_2}), \xi_i)]$$
$$= \mathbb{E}_{\mathbf{A}, \xi_i \in D_{m_1}, \xi \sim \mathbb{D}_1}[f(\mathbf{A}(\xi, \xi_2, .., \xi_{i-1}, \xi_{i+1}, ...\xi_{m_1}, D_{m_2}), \xi_i) - f(\mathbf{A}(D_{m_1}, D_{m_2}), \xi_i)]$$
$$= \mathbb{E}_{\mathbf{A}, \xi_i \in D_{m_1}, \xi \sim \mathbb{D}_1}[f(\mathbf{A}(D'_{m_1}, D_{m_2}), \xi_i) - f(\mathbf{A}(D_{m_1}, D_{m_2}), \xi_i)] \leq \beta,$$

where $D'_{m_1}$ and $D_{m_1}$ differ in at most one sample $\xi_i$. $\qquad\square$

*Proof of Part (b).* Similarly, we have

$$\mathbb{E}_\mathbf{A}[f(\mathbf{A}(D_{m_1}, D_{m_2}), \mathbb{D}_1) - f(\mathbf{A}(D_{m_1}, D_{m_2}), D_{m_1})]$$
$$= \mathbb{E}_{\mathbf{A}, \xi_i D_{m_1}, \xi \sim \mathbb{D}_1}[f(\mathbf{A}(D_{m_1}, D_{m_2}), \xi) - f(\mathbf{A}(D_{m_1}, D_{m_2}), \xi_i)]$$
$$= \mathbb{E}_{\mathbf{A}, \xi_i \in D_{m_1}, \xi \sim \mathbb{D}_1}[f(\mathbf{A}(\xi, \xi_2, .., \xi_{i-1}, \xi_{i+1}, ...\xi_{m_1}, D_{m_2}), \xi_i) - f(\mathbf{A}(D_{m_1}, D_{m_2}), \xi_i)]$$
$$= \mathbb{E}_{\mathbf{A}, \xi_i \in D_{m_1}, \xi \sim \mathbb{D}_1}[f(\mathbf{A}(D'_{m_1}, D_{m_2}), \xi_i) - f(\mathbf{A}(D_{m_1}, D_{m_2}), \xi_i)]$$
$$\leq \mathbb{E}_{\mathbf{A}, \xi_i \in D_{m_1}, \xi \sim \mathbb{D}_1}[L_f \|\mathbf{A}(D'_{m_1}, D_{m_2}) - \mathbf{A}(D_{m_1}, D_{m_2})\| \leq L_f \beta.$$

$\qquad\square$

To prove high probability bounds, we need the following lemma on the concentration behavior on the summation of weakly dependent random variables.

**Lemma 7** (Bousquet et al. 2020). *Let $Z = (Z_1, \ldots, Z_n)$ be a vector of independent random variables with each taking values in $\mathcal{Z}$, and $g_1, \ldots, g_n$ be some functions $g_i : \mathcal{Z}^n \to \mathbb{R}$ such that the following holds for any $i \in [n]$ :*

- *$|\mathbb{E}[g_i(Z) \mid Z_i]| \leq M$ a.s.,*

- *$\mathbb{E}[g_i(Z) \mid Z_{[n]\setminus\{i\}}] = 0$ a.s.,*

- *$g_i$ has a bounded difference $\beta$ with respect to all variables except for the $i$-th variable.*

*Then, for any $p \geq 2$,*

$$\left\| \sum_{i=1}^n g_i(Z) \right\|_p \leq 12\sqrt{2}pn\beta \lceil \log_2 n \rceil + 4M\sqrt{pn},$$

*where the $L_p$-norm of a random variable $Z$ is denoted by $\|Z\|_p := (\mathbb{E}[|Z|^p])^{1/p}$, $p \geq 1$.*

Next, we state the following well-known relationship between tail bounds and moment bounds.

**Lemma 8** (Bousquet et al. 2020; Vershynin 2018). *Let $a, b \in \mathbb{R}_+$. Let $Z$ be a random variable with $\|Z\|_p \leq \sqrt{p}a + pb$ and $p \geq 2$. Then, for any $\delta \in (0,1)$, we have, with probability at least $1 - \delta$*

$$|Z| \leq e\Big(a\sqrt{\log(\frac{e}{\delta})} + b\log(\frac{e}{\delta})\Big).$$

*Proof of Part (c).* In order to make use of Lemma 7 to obtain the generalization bounds, we will introduce:
$$h_i = \mathbb{E}_{\xi_i' \sim \mathbb{D}_1}[\mathbb{E}_{\xi_i \sim \mathbb{D}_1}[f(\mathbf{A}(D_{m_1}^i, D_{m_2}; \xi))] - f(\mathbf{A}(D_{m_1}^i, D_{m_2}; \xi_i)],$$

where $D_{m_1}^i = \{\xi_1, \xi_2, ..., \xi_{i-1}, \xi_i', \xi_{i+1}, ..., \xi_{m_1}\}$, and $\xi_i'$ obeys identical distribution of $\xi_i$.

Hence, we have:

$$|R(\mathbf{A}(D_{m_1}, D_{m_2}); \mathbb{D}_1) - R_s(\mathbf{A}(D_{m_1}, D_{m_2}); D_{m_1})|$$

$$= \frac{1}{m_1}\Big|\sum_{i=1}^{m_1} \big(\mathbb{E}_{\xi \sim \mathbb{D}_1} f(\mathbf{A}(D_{m_1}, D_{m_2}); \xi) - f(\mathbf{A}(D_{m_1}, D_{m_2}); \xi_i)\big)\Big|$$

$$\leq \frac{1}{m_1}\Big|\sum_{i=1}^{m_1} \big(\mathbb{E}_{\xi \sim \mathbb{D}_1} f(\mathbf{A}(D_{m_1}, D_{m_2}); \xi) - \mathbb{E}_{\xi \sim \mathbb{D}_1, \xi_i' \sim \mathbb{D}_1} f(\mathbf{A}(D_{m_1}^i, D_{m_2}); \xi)\big)\Big|$$

$$+ \Big|\frac{1}{m_1}\sum_{i=1}^{m_1} \mathbb{E}_{\xi_i' \sim \mathbb{D}_1}\mathbb{E}_{\xi \sim \mathbb{D}_1}\big[f(\mathbf{A}(D_{m_1}^i, D_{m_2}); \xi)\big] - f(\mathbf{A}(D_{m_1}^i, D_{m_2}); \xi_i)\Big|$$

$$+ \frac{1}{m_1}\Big|\sum_{i=1}^{m_1} \big(\mathbb{E}_{\xi_i' \sim \mathbb{D}_1} f(\mathbf{A}(D_{m_1}^i, D_{m_2}); \xi_i) - f(\mathbf{A}(D_{m_1}, D_{m_2}); \xi_i)\big)\Big|.$$

It then follows from the definition of uniform stability that

$$|R(\mathbf{A}(D_{m_1}, D_{m_2}); \mathbb{D}_1) - R_s(\mathbf{A}(D_{m_1}, D_{m_2}); D_{m_1})|$$

$$\leq 2\beta + \Big|\frac{1}{m_1}\sum_{i=1}^{m_1} \mathbb{E}_{\xi_i' \sim \mathbb{D}_1}\mathbb{E}_{\xi \sim \mathbb{D}_1}\big[f(\mathbf{A}(D_{m_1}^i, D_{m_2}); \xi)\big] - f(\mathbf{A}(D_{m_1}^i, D_{m_2}; \xi_i))\Big|$$

$$= 2\beta + \frac{1}{m_1}\Big|\sum_{i=1}^{m_1} h_i\Big|.$$

Notice that all conditions of 7 hold. Thus, the following outcome can be derived for any $p \geq 2$:

$$\Big\|\sum_{i=1}^{m_1} h_i(\xi)\Big\|_p \leq 12\sqrt{2}pm_1\beta \lceil \log_2 m_1 \rceil + 4M\sqrt{pm_1}.$$

Combining Lemma 7 and Lemma 8 with $h_i$ defined above, we have the following inequality with probability $1 - \delta$:

$$\Big|\sum_{i=1}^{m_1} h_i(\xi)\Big| \leq e\Big(\frac{4M}{\sqrt{m_1}}\sqrt{\log\frac{e}{\delta}} + 12\sqrt{2}\beta\lceil \log_2 m_1 \rceil \sqrt{\log\frac{e}{\delta}}\Big).$$

The deviation bound now follows immediately:

$$|R(\mathbf{A}(D_{m_1}, D_{m_2}); \mathbb{D}_1) - R_s(\mathbf{A}(D_{m_1}, D_{m_2}); D_{m_1})|$$

$$\leq 2\beta + e\Big(\frac{4M}{\sqrt{m_1}}\sqrt{\log\frac{e}{\delta}} + 12\sqrt{2}\beta\lceil \log_2 m_1 \rceil \sqrt{\log\frac{e}{\delta}}\Big).$$

The proof is completed. $\square$

## C MAIN PROOF

### C.1 APPROXIMATE EXPANSIVITY OF UPDATE RULES

With step size $\alpha_x$ and $\alpha_y$, the update rules for single-timescale can be presented:

$$G_s \left( \begin{bmatrix} \mathbf{x} \\ \mathbf{y} \end{bmatrix} \right) := \begin{bmatrix} \mathbf{x} - \alpha_x \nabla f(\mathbf{x}, \mathbf{y}) \\ \mathbf{y} - \alpha_y \nabla_{\mathbf{y}} g(\mathbf{x}, \mathbf{y}) \end{bmatrix}.$$

**Definition 5** (expansivity). An update rule is $\eta$-expansive if for every $x, x' \in \mathbb{R}^{d_1}$, $y, y' \in \mathbb{R}^{d_2}$:

$$\| G(\mathbf{x}, \mathbf{y}) - G(\mathbf{x}', \mathbf{y}') \|_2 \leq \eta \sqrt{\| \mathbf{x} - \mathbf{x}' \|_2^2 + \| \mathbf{y} - \mathbf{y}' \|_2^2}.$$

**Lemma 9.** *Suppose that Assumptions 1 and 2 hold for Problem (1). Then:*

1. *If $f$ and $g$ are non-convex functions, then $G_s$ is $(1 + \max\{l_f \alpha_x, l_g \alpha_y\})$-expansive with step size $\alpha_x$, $\alpha_y$.*

2. *If $f$ and $g$ are convex functions, then $G_s$ is $(\sqrt{2 + 2 \max\{(l_f \alpha_x)^2, (l_g \alpha_y)^2\}})$-expansive with step size $\alpha_x$, $\alpha_y$.*

3. *If $f$ and $g$ are strongly-convex with $\mu_f$ and $\mu_g$ respectively, then $G_s$ is $\sqrt{2(1 - 2\alpha_x(\mu_f + \mu_g) + \alpha_x^2 l^2)}$-expansive with step size:*

$$\frac{(u_f + \mu_g) - \sqrt{(u_f + \mu_g)^2 - 0.5 l^2}}{l^2} \leq \alpha_x = \alpha_y$$

$$\leq \min \left\{ \frac{1}{\mu_f + \mu_g}, \frac{(u_f + \mu_g) + \sqrt{(u_f + \mu_g)^2 - 0.5 l^2}}{l^2} \right\}.$$

*Proof.* In Case 1 with the NC-NC objectives and the smoothness of objectives on Assumptions 1 and 2, we have

$$\left\| G_s \left( \begin{bmatrix} \mathbf{x} \\ \mathbf{y} \end{bmatrix} \right) - G_s \left( \begin{bmatrix} \mathbf{x}' \\ \mathbf{y}' \end{bmatrix} \right) \right\| = \left\| \begin{bmatrix} \mathbf{x} - \mathbf{x}' - \alpha_x (\nabla f(\mathbf{x}, \mathbf{y}) - \nabla f(\mathbf{x}', \mathbf{y}')) \\ \mathbf{y} - \mathbf{y}' + \alpha_y (\nabla_{\mathbf{y}} g(\mathbf{x}, \mathbf{y}) - \nabla_{\mathbf{y}} g(\mathbf{x}', \mathbf{y}')) \end{bmatrix} \right\|$$

$$\leq \left\| \begin{bmatrix} \mathbf{x} - \mathbf{x}' \\ \mathbf{y} - \mathbf{y}' \end{bmatrix} \right\| + \left\| \begin{bmatrix} \alpha_x (\nabla f(\mathbf{x}, \mathbf{y}) - \nabla f(\mathbf{x}', \mathbf{y}')) \\ \alpha_y (\nabla_{\mathbf{y}} g(\mathbf{x}, \mathbf{y}) - \nabla_{\mathbf{y}} g(\mathbf{x}', \mathbf{y}')) \end{bmatrix} \right\|$$

$$\leq (1 + \max\{l_f \alpha_x, l_g \alpha_y\}) \left\| \begin{bmatrix} \mathbf{x} - \mathbf{x}' \\ \mathbf{y} - \mathbf{y}' \end{bmatrix} \right\|.$$

In case 2, with the monotonicity of the convex objective's gradient, we have:

$$\langle \mathbf{x} - \mathbf{x}', \alpha_x (\nabla f(\mathbf{x}, \mathbf{y}) - \nabla f(\mathbf{x}', \mathbf{y})) \rangle \geq 0$$
$$\langle \mathbf{y} - \mathbf{y}', \alpha_y (\nabla_{\mathbf{y}} g(\mathbf{x}', \mathbf{y}) - \nabla_{\mathbf{y}} g(\mathbf{x}', \mathbf{y}')) \rangle \geq 0.$$

Thus, the stated result then follows:

$$\left\| G_s \left( \begin{bmatrix} \mathbf{x} \\ \mathbf{y} \end{bmatrix} \right) - G_s \left( \begin{bmatrix} \mathbf{x}' \\ \mathbf{y} \end{bmatrix} \right) \right\|^2 = \left\| \begin{bmatrix} \mathbf{x} - \mathbf{x}' \\ \mathbf{y} - \mathbf{y} \end{bmatrix} \right\|^2 - 2 \begin{bmatrix} \mathbf{x} - \mathbf{x}' \\ \mathbf{y} - \mathbf{y} \end{bmatrix}^T \begin{bmatrix} \alpha_x (\nabla f(\mathbf{x}, \mathbf{y}) - \nabla f(\mathbf{x}', \mathbf{y})) \\ \alpha_y (\nabla_{\mathbf{y}} g(\mathbf{x}, \mathbf{y}) - \nabla_{\mathbf{y}} g(\mathbf{x}', \mathbf{y}))] \end{bmatrix}$$

$$+ \left\| \begin{bmatrix} \alpha_x (\nabla f(\mathbf{x}, \mathbf{y}) - \nabla f(\mathbf{x}', \mathbf{y})) \\ \alpha_y (\nabla_{\mathbf{y}} g(\mathbf{x}, \mathbf{y}) - \nabla_{\mathbf{y}} g(\mathbf{x}', \mathbf{y})) \end{bmatrix} \right\|^2$$

$$\leq \max\{(l_f \alpha_x)^2, (l_g \alpha_y)^2\} \left\| \begin{bmatrix} \mathbf{x} - \mathbf{x}' \\ \mathbf{y} - \mathbf{y} \end{bmatrix} \right\|^2 + \| \mathbf{x} - \mathbf{x}' \|^2.$$

$$(4)$$

and

$$\left\| G_s\left(\left[\begin{array}{c}\mathbf{x}'\\\mathbf{y}\end{array}\right]\right)-G_s\left(\left[\begin{array}{c}\mathbf{x}'\\\mathbf{y}'\end{array}\right]\right)\right\|^2=\left\|\left[\begin{array}{c}\mathbf{x}'-\mathbf{x}'\\\mathbf{y}-\mathbf{y}'\end{array}\right]\right\|^2-2\left[\begin{array}{c}\mathbf{x}'-\mathbf{x}'\\\mathbf{y}-\mathbf{y}'\end{array}\right]^T\left[\begin{array}{c}\alpha_x\left(\nabla f(\mathbf{x}',\mathbf{y}')-\nabla f\left(\mathbf{x}',\mathbf{y}\right)\right)\\\alpha_y\left(\nabla_\mathbf{y}g(\mathbf{x}',\mathbf{y}')-\nabla_\mathbf{y}g\left(\mathbf{x}',\mathbf{y}\right)\right)]\end{array}\right]$$

$$+\left\|\left[\begin{array}{c}\alpha_x\left(\nabla f(\mathbf{x}',\mathbf{y})-\nabla f\left(\mathbf{x}',\mathbf{y}'\right)\right)\\\alpha_y\left(\nabla_\mathbf{y}g(\mathbf{x}',\mathbf{y})-\nabla_\mathbf{y}g\left(\mathbf{x}',\mathbf{y}'\right)\right)\end{array}\right]\right\|^2$$

$$\leq\max\{(l_f\alpha_x)^2,(l_g\alpha_y)^2\}\left\|\left[\begin{array}{c}\mathbf{x}'-\mathbf{x}'\\\mathbf{y}-\mathbf{y}'\end{array}\right]\right\|^2+\|\mathbf{y}-\mathbf{y}'\|^2.$$

(5)

Combining the above equations 6, 7 and inequality $(\sum_{i=1}^k a_k)^2\leq k\sum_{i=1}^k a_k^2$, we can derive the expansive of update rule $G_s$ under convexity condition:

$$\left\|G_s\left(\left[\begin{array}{c}\mathbf{x}\\\mathbf{y}\end{array}\right]\right)-G_s\left(\left[\begin{array}{c}\mathbf{x}'\\\mathbf{y}'\end{array}\right]\right)\right\|^2\leq(2+2\max\{(l_f\alpha_x)^2,(l_g\alpha_y)^2\})\left\|\left[\begin{array}{c}\mathbf{x}-\mathbf{x}'\\\mathbf{y}-\mathbf{y}'\end{array}\right]\right\|^2.$$

If $f$ and $g$ are strongly-convex, then, $\widetilde{f}(\mathbf{x},\mathbf{y})=f(\mathbf{x},\mathbf{y})-\frac{\mu_f}{2}(\|\mathbf{x}\|^2+\|\mathbf{y}\|^2)$ and $\widetilde{g}(\mathbf{x},\mathbf{y})=g(\mathbf{x},\mathbf{y})-\frac{\mu_g}{2}(\|\mathbf{x}\|^2+\|\mathbf{y}\|^2)$ will be convex. With the above conclusions, we can derive the following:

$$\left\|G_T\left(\left[\begin{array}{c}\mathbf{x}\\\mathbf{y}\end{array}\right]\right)-G_s\left(\left[\begin{array}{c}\mathbf{x}'\\\mathbf{y}\end{array}\right]\right)\right\|^2$$

$$=\left\|\left[\begin{array}{c}\mathbf{x}-\mathbf{x}'\\\mathbf{y}-\mathbf{y}\end{array}\right]\right\|^2-2\alpha_x\left[\begin{array}{c}\mathbf{x}-\mathbf{x}'\\\mathbf{y}-\mathbf{y}\end{array}\right]^T\left[\begin{array}{c}(\nabla f(\mathbf{x},\mathbf{y})-\nabla f\left(\mathbf{x}',\mathbf{y}\right))\\(\nabla_\mathbf{y}g(\mathbf{x},\mathbf{y})-\nabla_\mathbf{y}g\left(\mathbf{x}',\mathbf{y}\right))\end{array}\right]$$

$$+\alpha_x{}^2\left\|\left[\begin{array}{c}(\nabla f(\mathbf{x},\mathbf{y})-\nabla f\left(\mathbf{x}',\mathbf{y}\right))\\(\nabla_\mathbf{y}g(\mathbf{x},\mathbf{y})-\nabla_\mathbf{y}g\left(\mathbf{x}',\mathbf{y}\right))\end{array}\right]\right\|^2$$

$$=(1-(\alpha_x\mu_f+\alpha_x\mu_g))^2\left\|\left[\begin{array}{c}\mathbf{x}-\mathbf{x}'\\\mathbf{y}-\mathbf{y}\end{array}\right]\right\|^2+\alpha_x{}^2\left\|\left[\begin{array}{c}(\nabla\widetilde{f}(\mathbf{x},\mathbf{y})-\nabla\widetilde{f}\left(\mathbf{x}',\mathbf{y}\right))\\(\nabla_\mathbf{y}\widetilde{g}(\mathbf{x},\mathbf{y})-\nabla_\mathbf{y}\widetilde{g}\left(\mathbf{x}',\mathbf{y}\right))\end{array}\right]\right\|^2$$

$$-2\left(1-\alpha_x\mu_f-\alpha_x\mu_g\right)\alpha_x\left[\begin{array}{c}\mathbf{x}-\mathbf{x}'\\\mathbf{y}-\mathbf{y}\end{array}\right]^T\left[\begin{array}{c}\left(\nabla\widetilde{f}(\mathbf{x},\mathbf{y})-\nabla\widetilde{f}\left(\mathbf{x}',\mathbf{y}\right)\right)\\(\nabla_\mathbf{y}\widetilde{g}(\mathbf{x},\mathbf{y})-\nabla_\mathbf{y}\widetilde{g}\left(\mathbf{x}',\mathbf{y}\right))\end{array}\right]$$

$$\leq\left(1-2\alpha_x\left(\mu_f+\mu_g\right)+\alpha_x{}^2l^2\right)\|\mathbf{x}-\mathbf{x}'\|^2.$$

The penultimate inequality arises from the smoothness of $\widetilde{f},\widetilde{g}$, which is based on our assumption for simplicity that $l=\max\{l_f,l_g\}$, and the details will be revealed as follows:

$$l^2\left\|\left[\begin{array}{c}\mathbf{x}-\mathbf{x}'\\\mathbf{y}-\mathbf{y}\end{array}\right]\right\|^2\geq\left\|\left[\begin{array}{c}\nabla f(\mathbf{x},\mathbf{y})-\nabla f\left(\mathbf{x}',\mathbf{y}\right)\\\nabla_\mathbf{y}g(\mathbf{x},\mathbf{y})-\nabla_\mathbf{y}g\left(\mathbf{x}',\mathbf{y}\right)\end{array}\right]\right\|^2$$

$$=\left\|\left[\begin{array}{c}\left(\nabla\widetilde{f}(\mathbf{x},\mathbf{y})-\nabla\widetilde{f}\left(\mathbf{x}',\mathbf{y}\right)\right)\\(\nabla_\mathbf{y}\widetilde{g}(\mathbf{x},\mathbf{y})-\nabla_\mathbf{y}\widetilde{g}\left(\mathbf{x}',\mathbf{y}\right))\end{array}\right]\right\|^2+(\mu_f+\mu_g)^2\left\|\left[\begin{array}{c}\mathbf{x}-\mathbf{x}'\\\mathbf{y}-\mathbf{y}\end{array}\right]\right\|^2$$

$$+2\left(\mu_f+\mu_g\right)\left[\begin{array}{c}\mathbf{x}-\mathbf{x}'\\\mathbf{y}-\mathbf{y}\end{array}\right]^T\left[\begin{array}{c}\left(\nabla\widetilde{f}(\mathbf{x},\mathbf{y})-\nabla\widetilde{f}\left(\mathbf{x}',\mathbf{y}\right)\right)\\(\nabla_\mathbf{y}\widetilde{g}(\mathbf{x},\mathbf{y})-\nabla_\mathbf{y}\widetilde{g}\left(\mathbf{x}',\mathbf{y}\right))\end{array}\right]$$

$$\geq\left\|\left[\begin{array}{c}\left(\nabla\widetilde{f}(\mathbf{x},\mathbf{y})-\nabla\widetilde{f}\left(\mathbf{x}',\mathbf{y}\right)\right)\\(\nabla_\mathbf{y}\widetilde{g}(\mathbf{x},\mathbf{y})-\nabla_\mathbf{y}\widetilde{g}\left(\mathbf{x}',\mathbf{y}\right))\end{array}\right]\right\|^2+(\mu_f+\mu_g)^2\left\|\left[\begin{array}{c}\mathbf{x}-\mathbf{x}'\\\mathbf{y}-\mathbf{y}\end{array}\right]\right\|^2.$$

Similar to the convex case, we can have:

$$\left\|G_T\left(\left[\begin{array}{c}\mathbf{x}\\\mathbf{y}\end{array}\right]\right)-G_s\left(\left[\begin{array}{c}\mathbf{x}'\\\mathbf{y}'\end{array}\right]\right)\right\|^2\leq2\left(1-2\alpha_x\left(\mu_f+\mu_g\right)+\alpha_x{}^2l^2\right)\left\|\left[\begin{array}{c}\mathbf{x}-\mathbf{x}'\\\mathbf{y}-\mathbf{y}'\end{array}\right]\right\|^2.$$

$\square$

## C.2 SINGLE TIMESCALE

We first introduce the following lemma before providing the proof of the Theorem.

**Lemma 10** (Hardt et al. (2016)). *Consider two sequences of updates $G_s^1, ..., G_s^K$ and $(G_s^1)', ..., (G_s^K)'$ with initial points $\mathbf{x}_0 = \mathbf{x}_0'$, $\mathbf{y}_0 = \mathbf{y}_0'$. Define $\delta_k = \sqrt{\|\mathbf{x}_k - \mathbf{x}_k'\|^2 + \|\mathbf{y}_k - \mathbf{y}_k'\|^2}$. Then, we have:*

$$
\delta_{k+1} \leq
\begin{cases}
\eta \delta_k & \text{if } G_s^k = (G_s^k)' \text{ is } \eta\text{-expansive} \\
\min(\eta, 1)\delta_k + 2\sigma & \text{if } \sup \left\| \begin{bmatrix} \mathbf{x} \\ \mathbf{y} \end{bmatrix} - G\left(\begin{bmatrix} \mathbf{x} \\ \mathbf{y} \end{bmatrix}\right) \right\| \leq \sigma \\
& G_s^k \text{ is } \eta \text{ expansive}
\end{cases}
$$

*Proof.* The first part of the inequality is obvious from the definition of expansivity and the assumption of $G_s^k = (G_s^k)'$. For the second bound, note that:

$$
\begin{aligned}
\delta_{k+1} &= \left\| G_s\left(\begin{bmatrix} \mathbf{x}_k \\ \mathbf{y}_k \end{bmatrix}\right) - G_s'\left(\begin{bmatrix} \mathbf{x}_k' \\ \mathbf{y}_k' \end{bmatrix}\right) \right\| \\
&\leq \left\| G_s\left(\begin{bmatrix} \mathbf{x}_k \\ \mathbf{y}_k \end{bmatrix}\right) - \begin{bmatrix} \mathbf{x}_k \\ \mathbf{y}_k \end{bmatrix} + \begin{bmatrix} \mathbf{x}_k' \\ \mathbf{y}_k' \end{bmatrix} - G_s'\left(\begin{bmatrix} \mathbf{x}_k' \\ \mathbf{y}_k' \end{bmatrix}\right) \right\| + \left\| \begin{bmatrix} \mathbf{x}_k - \mathbf{x}_k' \\ \mathbf{y}_k - \mathbf{y}_k' \end{bmatrix} \right\| \\
&\leq \delta_k + \left\| G_s\left(\begin{bmatrix} \mathbf{x}_k \\ \mathbf{y}_k \end{bmatrix}\right) - \begin{bmatrix} \mathbf{x}_k \\ \mathbf{y}_k \end{bmatrix} \right\| + \left\| G_s'\left(\begin{bmatrix} \mathbf{x}_k' \\ \mathbf{y}_k' \end{bmatrix}\right) - \begin{bmatrix} \mathbf{x}_k' \\ \mathbf{y}_k' \end{bmatrix} \right\| \\
&\leq \delta_k + 2\sigma.
\end{aligned}
$$

Also, $\delta_{k+1}$ can be further expressed as:

$$
\begin{aligned}
\delta_{k+1} &= \left\| G_s\left(\begin{bmatrix} \mathbf{x}_k \\ \mathbf{y}_k \end{bmatrix}\right) - G_s'\left(\begin{bmatrix} \mathbf{x}_k' \\ \mathbf{y}_k' \end{bmatrix}\right) \right\| \\
&\leq \left\| G_s\left(\begin{bmatrix} \mathbf{x}_k \\ \mathbf{y}_k \end{bmatrix}\right) - G_s\left(\begin{bmatrix} \mathbf{x}_k' \\ \mathbf{y}_k' \end{bmatrix}\right) + G_s\left(\begin{bmatrix} \mathbf{x}_k' \\ \mathbf{y}_k' \end{bmatrix}\right) - G_s'\left(\begin{bmatrix} \mathbf{x}_k' \\ \mathbf{y}_k' \end{bmatrix}\right) \right\| \\
&\leq \left\| G_s\left(\begin{bmatrix} \mathbf{x}_k \\ \mathbf{y}_k \end{bmatrix}\right) - G_s\left(\begin{bmatrix} \mathbf{x}_k' \\ \mathbf{y}_k' \end{bmatrix}\right) \right\| + \left\| \begin{bmatrix} \mathbf{x}_k' \\ \mathbf{y}_k' \end{bmatrix} - G_s\left(\begin{bmatrix} \mathbf{x}_k' \\ \mathbf{y}_k' \end{bmatrix}\right) \right\| + \left\| \begin{bmatrix} \mathbf{x}_k' \\ \mathbf{y}_k' \end{bmatrix} - G_s'\left(\begin{bmatrix} \mathbf{x}_k' \\ y_{k'} \end{bmatrix}\right) \right\| \\
&\leq \eta \delta_k + 2\sigma.
\end{aligned}
$$

Combining the above completes the proof of the Lemma 10. $\square$

Now, we are ready to prove Theorem 2:

*Proof of Part(a).* Suppose that $D_{m_1}$ and $D_{m_1}'$ are two neighboring sets differing only in one sample. Consider the updates $G_s^1, ..., G_s^K$ and $(G_s^1)', ..., (G_s^K)'$. We can observe that the example chosen by the algorithm is the same in $D_{m_1}, D_{m_1}'$ at step $k$ with probability $1 - 1/m_1$ and different with probability $1/m_1$. In the former case, we have identical update rules, while $\sqrt{1 - 2\alpha_x (\mu_f + \mu_g) + \alpha_x^2 l^2}$-

expansive can be employed in the latter through lemma 10.

$$\mathbb{E}\left[\delta_{k+1}\right] \leq \left(1 - \frac{1}{m_1}\right)\left(2\left(1 - 2\alpha_x\left(\mu_f + \mu_g\right) + \alpha_x^2 l^2\right)\right)^{1/2}\mathbb{E}\left[\delta_k\right] + \frac{1}{m_1}\mathbb{E}\left[\delta_k\right] + \frac{1}{m_1}2\sqrt{(\alpha_x L_f)^2 + (\alpha_x L_g)^2}$$

$$\leq \left(2\left(1 - 2\alpha_x\left(\mu_f + \mu_g\right) + \alpha_x^2 l^2\right)\right)^{1/2}\mathbb{E}\left[\delta_k\right] + \frac{2}{m_1}\sqrt{(\alpha_x L_f)^2 + (\alpha_x L_g)^2}$$

$$\leq \frac{2\sqrt{(\alpha_x L_f)^2 + (\alpha_x L_g)^2}}{m_1}\sum_{i=0}^{k}\left(2\left(1 - 2\alpha_x\left(\mu_y + \mu_g\right) + \alpha_x^2 l^2\right)\right)^{i/2}$$

$$\leq \frac{2\sqrt{(\alpha_x L_f)^2 + (\alpha_x L_g)^2}}{m_1}\sum_{i=0}^{\infty}\left(2\left(1 - 2\alpha_x\left(\mu_f + \mu_g\right) + \alpha_x^2 l^2\right)\right)^{i/2}$$

$$(1) \leq \frac{2\sqrt{(\alpha_x L_f)^2 + (\alpha_x L_g)^2}}{m_1}\sum_{i=0}^{\infty}\left(1 - 2\alpha_x\left(\mu_f + \mu_g\right) + \alpha_x^2 l^2 + 0.5\right)^{i}$$

$$= \frac{\sqrt{(\alpha_x L_f)^2 + (\alpha_x L_g)^2}}{m_1\left(\alpha_x\left(\mu_f + \mu_g\right) - \frac{\alpha_x^2 l^2}{2} + 0.25\right)}$$

$$= \frac{\sqrt{L_f^2 + L_g^2}}{m_1\left(\mu_f + \mu_g - (\alpha_x l)^2/2 + 0.25\right)}.$$

Here (1) comes from the mean equality $\sqrt{ab} \leq (a+b)/2$ for any $a, b \geq 0$ and the assumption of $\frac{(u_f + \mu_g) - \sqrt{(u_f + \mu_g)^2 - 0.5l^2}}{l^2} \leq \alpha_x \leq \frac{(u_f + \mu_g) + \sqrt{(u_f + \mu_g)^2 - 0.5l^2}}{l^2}$, which finishes the proof. $\qquad\square$

*Proof of Part(b).* The proof of Part(b) is analogous to the above, thus we use the same notations for this part.

$$\mathbb{E}\left[\delta_{k+1}\right] \leq \left(1 - \frac{1}{m_1}\right)\left(2 + 2\max\left\{l_f^2\alpha_x^2, l_y^2\alpha_y^2\right\}\right)^{1/2}\mathbb{E}\left[\delta_k\right] + \frac{1}{m_1}\mathbb{E}\left[\delta_k\right] + \frac{2}{m_1}\sqrt{L_f^2\alpha_x^2 + L_g^2\alpha_y^2}$$

$$= \left(2 + 2\max\left\{l_f^2\alpha_x^2, l_g^2\alpha_y^2\right\}\right)^{1/2}\mathbb{E}\left[\delta_k\right] + \frac{2\sqrt{L_f^2\alpha_x^2 + L_g^2\alpha_y^2}}{m_1}$$

$$\mathbb{E}\left[\delta_k\right] \leq \frac{2\sqrt{L_f^2\alpha_x^2 + L_g^2\alpha_y^2}}{m_1} \cdot \frac{\left(2 + 2\max\left\{l_f^2\alpha_x^2, l_g^2\alpha_y^2\right\}\right)^{\frac{k+1}{2}} - 1}{\sqrt{2 + 2\max\left\{l_f^2\alpha_x^2, l_g^2\alpha_y^2\right\}} - 1}$$

$$\mathbb{E}\left[\delta_k\right] \leq \mathcal{O}\left(\frac{\sqrt{L_f^2\alpha_x^2 + L_g^2\alpha_y^2}\left(2 + 2\max\left\{l_f^2\alpha_x^2, l_g^2\alpha_y^2\right\}\right)^{\frac{k+1}{2}}}{m_1}\right).$$

$$\square$$

To prove stability in the NC-NC case, we introduce the following lemma:

**Lemma 11** (Hardt et al. (2016))**.** *Assume that $f(\mathbf{x}, \mathbf{y}; \xi)$ is $L_f$-Lipschitz continuous and $0 \leq f(\mathbf{x}, \mathbf{y}; \xi) \leq 1$. Let $D_{m_1}$ and $D'_{m_1}$ be two datasets differing in only one sample. Denote $(\mathbf{x}_K, \mathbf{y}_K)$ and $(\mathbf{x}'_K, \mathbf{y}'_K)$ as the output of $K$ steps of SSGD (single-timescale algorithm) on $D_{m_1}$ and $D'_{m_1}$, respectively. Then, the following holds for every $k \in \{0, 1, ..., K\}$, where $\delta_k = \sqrt{\|\mathbf{x}_k - \mathbf{x}'_k\|^2 + \|\mathbf{y}_k - \mathbf{y}'_k\|^2}$:*

$$\mathbb{E}\left[|f\left(\mathbf{x}_k, \mathbf{y}_k; \xi\right) - f\left(\mathbf{x}'_k, \mathbf{y}'_k; \xi\right)|\right] \leq \frac{k_0}{m_1} + L_f\mathbb{E}\left[\delta_k \mid \delta_{k_0} = 0\right].$$

*Proof of Part(c).* Applying Lemma 11, we get ready to prove the NC-NC case. Analogous to the previous case, we have:

$$\mathbb{E}\left[\delta_{k+1}\right] \le \left(1 - \frac{1}{m_1}\right)\left(1 + \frac{cl}{k}\right)\mathbb{E}\left[\delta_k\right] + \frac{1}{m}\left(1 + \frac{cl}{k}\right)\mathbb{E}\left[\delta_k\right] + \frac{2c\sqrt{l_f^2 + l_g^2}}{k}$$

$$= \left(1 + \frac{cl}{k}\right)\mathbb{E}\left[\delta_k\right] + \frac{2c\sqrt{l_f^2 + l_g^2}}{m_1 k}.$$

The following can be derived:

$$\mathbb{E}\left[\delta_K \mid \delta_{k_0} = 0\right] \le \sum_{k=k_0+1}^{K} \prod_{t=k+1}^{T} \left(1 + \frac{cl}{t}\right)\frac{2c\sqrt{l_f^2 + l_g^2}}{m_1 k}$$

$$\le \sum_{k=k_0+1}^{K} \prod_{t=k+1}^{T} \left\{\exp\left(\frac{cl}{t}\right)\right\}\frac{2c\sqrt{l_f^2 + l_g^2}}{m_1 k}$$

$$\le \sum_{k=k_0+1}^{K} \exp\left(\sum_{t=k+1}^{K} \frac{cl}{t}\right)\frac{2c\sqrt{l_f^2 + l_g^2}}{m_1 k}$$

$$\le \sum_{k=k_0+1}^{k} \exp(cl \cdot \log(K/k))\frac{2c\sqrt{l_f^2 + l_g^2}}{m_1 k}$$

$$\le \frac{2c\sqrt{l_f^2 + l_g^2}}{m_1} \sum_{k=k_0+1}^{K} k^{-cl-1}$$

$$\le \frac{2\sqrt{l_f^2 + l_g^2}}{m_1 l}\left(\frac{K}{k_0}\right)^{cl}.$$

Hence, Lemma 11 indicates:

$$\mathbb{E}\left[|f(x,y) - f(x',y')|\right] \le \frac{k_0}{m_1} + \frac{2L_f\sqrt{l_f^2 + l_g^2}}{m_1 l}\left(\frac{K}{k_0}\right)^{cl}.$$

The right hand side is approximately minimized when

$$k_0 = \left(2cL_f\sqrt{l_f^2 + l_g^2}\right)^{\frac{1}{cl+1}} \cdot K^{\frac{cl}{cl+1}}.$$

Therefore, we have

$$\beta \le \mathcal{O}\left(\frac{\left(2cL_f\sqrt{l_f^2 + l_g^2}\right)^{\frac{1}{cl+1}} \cdot K^{\frac{cl}{cl+1}}}{m_1 cl}\right)$$

for argument stability.

$\square$

## C.3 TWO-TIMESCALE SGD (TSGD)

### C.3.1 STANDARD SETTINGS

With step size $\alpha_x$ and $\alpha_y$, the update rule for two-timescale can be presented as:

$$G_T\left(\begin{bmatrix} \mathbf{x_k} \\ \mathbf{y_k} \end{bmatrix}\right) := \begin{bmatrix} \mathbf{x_k} - \alpha_x \nabla f(\mathbf{x}_k, \mathbf{y}_k^T) \\ \mathbf{y_k} - \alpha_y \sum_{t=1}^{T} \nabla_{\mathbf{y}} g(\mathbf{x}_k, \mathbf{y}_k^t) \end{bmatrix}.$$

Analogous to the single-timescale case, we first provide the expansivity of the update rules.

**Lemma 12.** *Suppose that Assumptions 1 and 2 hold for Problem (1). Let* $\alpha l = \max\{\alpha_x l_f, \frac{1+(\alpha_y l_g)^2}{1-\alpha_y l_g}\}$ *for simplicity sake and assume* $\alpha_y l_g \leq 1$. *Then:*

1. *If* $f$ *and* $g$ *are non-convex functions,* $G_T$ *is* $(1 + \alpha lT)$*-expansive.*

2. *If* $f$ *and* $g$ *are convex functions,* $G_T$ *is* $(1 + \alpha l)$*-expansive with step size* $\alpha_x$, $\alpha_y$.

3. *If* $f$ *and* $g$ *are strongly-convex with* $\mu_f$ *and* $\mu_g$ *respectively,* $G_T$ *is* $1 + \alpha l$*-expansive with step size:*

$$\alpha_x = \alpha_y \leq \frac{1}{\mu_f + \mu_g}.$$

*Proof.* In Case 1 with the NC-NC objectives by the triangle inequality, we have:

$$\left\| G_T\left( \begin{bmatrix} \mathbf{x} \\ \mathbf{y} \end{bmatrix} \right) - G_T\left( \begin{bmatrix} \mathbf{x}' \\ \mathbf{y}' \end{bmatrix} \right) \right\| \leq \left\| G_T\left( \begin{bmatrix} \mathbf{x} \\ \mathbf{y} \end{bmatrix} \right) - G_T\left( \begin{bmatrix} \mathbf{x}' \\ \mathbf{y} \end{bmatrix} \right) \right\| + \left\| G_T\left( \begin{bmatrix} \mathbf{x}' \\ \mathbf{y} \end{bmatrix} \right) - G_T\left( \begin{bmatrix} \mathbf{x}' \\ \mathbf{y}' \end{bmatrix} \right) \right\|$$

The first item can be derived from:

$$\left\| G_T\left( \begin{bmatrix} \mathbf{x} \\ \mathbf{y} \end{bmatrix} \right) - G_T\left( \begin{bmatrix} \mathbf{x}' \\ \mathbf{y} \end{bmatrix} \right) \right\| = \left\| \begin{bmatrix} \mathbf{x} - \mathbf{x}' - \alpha_x \left( \nabla f(\mathbf{x}, \mathbf{y}) - \nabla f\left(\mathbf{x}', \mathbf{y}\right) \right) \\ \mathbf{y} - \mathbf{y} + \alpha_y \sum_{y=1}^{T} \left( \nabla_{\mathbf{y}} g(\mathbf{x}, \mathbf{y}^t) - \nabla_{\mathbf{y}} g\left(\mathbf{x}', \mathbf{y}^t\right) \right) \end{bmatrix} \right\|$$
$$\leq (1 + \alpha_y T l_g) \left\| \mathbf{x} - \mathbf{x}' \right\|$$

The second item can be derived from:

$$\left\| G_T\left( \begin{bmatrix} \mathbf{x}' \\ \mathbf{y} \end{bmatrix} \right) - G_T\left( \begin{bmatrix} \mathbf{x}' \\ \mathbf{y}' \end{bmatrix} \right) \right\| = \left\| \begin{bmatrix} \mathbf{x}' - \mathbf{x}' - \alpha_x \left( \nabla f(\mathbf{x}', \mathbf{y}) - \nabla f\left(\mathbf{x}', \mathbf{y}'\right) \right) \\ \mathbf{y} - \mathbf{y}' + \sum_{t=0}^{T-1} \alpha_y \left( \nabla_{\mathbf{y}} g(\mathbf{x}', \mathbf{y}^t) - \nabla_{\mathbf{y}} g\left(\mathbf{x}', \mathbf{y}^{\mathbf{t}'}\right) \right) \end{bmatrix} \right\|$$
$$\leq \left\| \begin{bmatrix} \mathbf{x}' - \mathbf{x}' \\ \mathbf{y} - \mathbf{y}' \end{bmatrix} \right\| + \left\| \begin{bmatrix} \alpha_x \left( \nabla f(\mathbf{x}', \mathbf{y}) - \nabla f\left(\mathbf{x}', \mathbf{y}'\right) \right) \\ \sum_{t=0}^{T-1} \alpha_y \left( \nabla_{\mathbf{y}} g(\mathbf{x}', \mathbf{y}^t) - \nabla_{\mathbf{y}} g\left(\mathbf{x}', \mathbf{y}^{\mathbf{t}'}\right) \right) \end{bmatrix} \right\|$$

From the Lipschitz continuous, we have:

$$\sum_{t=0}^{T-1} \alpha_y \left( \nabla_y g\left(\mathbf{x}, \mathbf{y}^t\right) - \nabla_y g\left(\mathbf{x}, \mathbf{y}^t\right) \right) \leq \sum_{t=0}^{T-1} \alpha_y l_g \left\| \mathbf{y}^t - \mathbf{y}^t \right\|$$

Now we consider the $t$-th update:

$$\alpha_y l_g \left\| \mathbf{y}^t - \mathbf{y}^t \right\| = \alpha_y l_g \left\| \mathbf{y}^{t-1} - \alpha_y \nabla_y g\left(\mathbf{x}', \mathbf{y}^{t-1}\right) - \mathbf{y}^{t-1} + \alpha_y \nabla_y g\left(\mathbf{x}', \mathbf{y}^{t-1}\right) \right\|$$
$$\leq \alpha_y l_g \left\| \mathbf{y}^{t-1} - \left(\mathbf{y}^{t-1}\right)' \right\| + (\alpha_y l_g)^2 \left\| \mathbf{y}^{t-1} - \left(\mathbf{y}^{t-1}\right)' \right\|$$
$$\cdots$$
$$\leq (\alpha_y l_g)^t \left\| \mathbf{y}^0 - \left(\mathbf{y}^0\right)' \right\| + (\alpha_y l_g)^{t+1} \left\| \mathbf{y}^0 - \left(\mathbf{y}^0\right)' \right\|$$

According to the accumulation of the both side, we have:

$$\sum_{t=0}^{T-1} \alpha_y l_g \left\| \mathbf{y}^t - \left(\mathbf{y}^t\right)' \right\| \leq \alpha_y l_g \left\| \mathbf{y}^0 - \left(\mathbf{y}^0\right)' \right\| \| \sum_{t=1}^{T-1} \left[ (\alpha_y l_g)^t \left\| \mathbf{y}^0 - \left(\mathbf{y}^0\right)' \right\| + (\alpha_y l_g)^{t+1} \left\| \mathbf{y}^0 - \left(\mathbf{y}^0\right)' \right\| \right] \|$$
$$= \left[ \frac{1 - (\alpha_y l g)^T}{1 - \alpha_y l g} + \frac{(\alpha_y l g)^2 - (\alpha_y l g)^{T+1}}{1 - \alpha_y l_g} \right] \left\| \mathbf{y}^0 - \left(\mathbf{y}^0\right)' \right\|$$
$$= \left[ \frac{1 - (\alpha_y l_g)^T + (\alpha_y l_g)^2 - (\alpha_y l_g)^{T+1}}{1 - \alpha_y l_g} \right] \left\| \mathbf{y}^0 - \left(\mathbf{y}^0\right)' \right\|$$
$$\leq \frac{1 + (\alpha_y l_g)^2}{1 - \alpha_y l_g} \left\| \mathbf{y} - \left(\mathbf{y}\right)' \right\|$$

Let $\alpha l = \max\{\alpha_y l_g, \frac{1+(\alpha_y l_g)^2}{1-\alpha_y l_g}\}$, then:

$$\left\| G_T\left(\begin{bmatrix} \mathbf{x} \\ \mathbf{y} \end{bmatrix}\right) - G_T\left(\begin{bmatrix} \mathbf{x}' \\ \mathbf{y}' \end{bmatrix}\right) \right\| \leq (1 + T\alpha l)\left\| \begin{bmatrix} \mathbf{x} - \mathbf{x}' \\ \mathbf{y} - \mathbf{y}' \end{bmatrix} \right\|.$$

In case 2, with the monotonicity of the convex objective's gradient, we have:

$$\langle \mathbf{x} - \mathbf{x}', \alpha_x \left(\nabla f(\mathbf{x}, \mathbf{y}) - \nabla f\left(\mathbf{x}', \mathbf{y}\right)\right)\rangle \geq 0$$
$$\langle \mathbf{y} - \mathbf{y}', \alpha_y \left(\nabla_{\mathbf{y}} g(\mathbf{x}', \mathbf{y}) - \nabla_{\mathbf{y}} g\left(\mathbf{x}', \mathbf{y}'\right)\right)\rangle \geq 0.$$

Thus, the stated result then follows:

$$\left\| G_T\left(\begin{bmatrix} \mathbf{x} \\ \mathbf{y} \end{bmatrix}\right) - G_T\left(\begin{bmatrix} \mathbf{x}' \\ \mathbf{y} \end{bmatrix}\right) \right\|^2 = \left\| \begin{bmatrix} \mathbf{x} - \mathbf{x}' \\ \mathbf{y} - \mathbf{y} \end{bmatrix} \right\|^2 - 2\begin{bmatrix} \mathbf{x} - \mathbf{x}' \\ \mathbf{y} - \mathbf{y} \end{bmatrix}^T \begin{bmatrix} \alpha_x \left(\nabla f(\mathbf{x}, \mathbf{y}) - \nabla f\left(\mathbf{x}', \mathbf{y}\right)\right) \\ \sum_{t=0}^{T-1} \alpha_y \left(\nabla_{\mathbf{y}} g(\mathbf{x}, \mathbf{y}^t) - \nabla_{\mathbf{y}} g\left(\mathbf{x}, \mathbf{y}^{\mathbf{t}'}\right)\right) \end{bmatrix}$$
$$+ \left\| \begin{bmatrix} \alpha_x \left(\nabla f(\mathbf{x}, \mathbf{y}) - \nabla f\left(\mathbf{x}', \mathbf{y}\right)\right) \\ \sum_{t=0}^{T-1} \alpha_y \left(\nabla_{\mathbf{y}} g(\mathbf{x}, \mathbf{y}^t) - \nabla_{\mathbf{y}} g\left(\mathbf{x}, \mathbf{y}^{\mathbf{t}'}\right)\right) \end{bmatrix} \right\|^2$$
$$\leq \max\left\{ (l_f \alpha_x)^2, \left(\frac{1+(\alpha_y l_g)^2}{1-\alpha_y l_g}\right)^2 \right\} \left\| \begin{bmatrix} \mathbf{x} - \mathbf{x}' \\ \mathbf{y} - \mathbf{y} \end{bmatrix} \right\|^2 + \|\mathbf{x} - \mathbf{x}'\|^2.$$
(6)

and the second decomposition can be obtained by the NC-NC case:

$$\left\| G_T\left(\begin{bmatrix} \mathbf{x}' \\ \mathbf{y} \end{bmatrix}\right) - G_T\left(\begin{bmatrix} \mathbf{x}' \\ \mathbf{y}' \end{bmatrix}\right) \right\| \leq \left(1 + \max\{l_f \alpha_x, \frac{1+(\alpha_y l_g)^2}{1-\alpha_y l_g}\}\right) \|\mathbf{y} - \mathbf{y}'\|. \quad (7)$$

let $\alpha l = \max\{\alpha_x l_f, \frac{1+(\alpha_y l_g)^2}{1-\alpha_y l_g}\}$. Combining the above equations 6, 7 and inequality $\sqrt{1 + (\alpha l)^2} \leq (1 + \alpha l)^2$, then we can derive the expansive of update rule $G_T$ under convexity condition:

$$\left\| G_T\left(\begin{bmatrix} \mathbf{x} \\ \mathbf{y} \end{bmatrix}\right) - G_T\left(\begin{bmatrix} \mathbf{x}' \\ \mathbf{y}' \end{bmatrix}\right) \right\| \leq (1 + \alpha l)\left\| \begin{bmatrix} \mathbf{x} - \mathbf{x}' \\ \mathbf{y} - \mathbf{y}' \end{bmatrix} \right\|.$$

If $f$ and $g$ are strongly-convex, then, $\widetilde{f}(\mathbf{x}, \mathbf{y}) = f(\mathbf{x}, \mathbf{y}) - \frac{\mu_f}{2}(\|\mathbf{x}\|^2 + \|\mathbf{y}\|^2)$ and $\widetilde{g}(\mathbf{x}, \mathbf{y}) = g(\mathbf{x}, \mathbf{y}) - \frac{\mu_g}{2}(\|\mathbf{x}\|^2 + \|\mathbf{y}\|^2)$ will be convex. Let $\alpha_x = \alpha_y = \alpha$ and denote $\alpha l = \max\{\alpha_x l_f, \frac{1+(\alpha_y l_g)^2}{1-\alpha_y l_g}\}$, we can derive the following with the conclusions from the convex case:

$$\left\| G_T\left(\begin{bmatrix} \mathbf{x} \\ \mathbf{y} \end{bmatrix}\right) - G_T\left(\begin{bmatrix} \mathbf{x}' \\ \mathbf{y} \end{bmatrix}\right) \right\|^2$$
$$= \left\| \begin{bmatrix} \mathbf{x} - \mathbf{x}' \\ \mathbf{y} - \mathbf{y} \end{bmatrix} \right\|^2 - 2\alpha_x \begin{bmatrix} \mathbf{x} - \mathbf{x}' \\ \mathbf{y} - \mathbf{y} \end{bmatrix}^T \begin{bmatrix} \left(\nabla f(\mathbf{x}, \mathbf{y}) - \nabla f\left(\mathbf{x}', \mathbf{y}\right)\right) \\ \sum_{t=0}^{T-1} \left(\nabla_{\mathbf{y}} g(\mathbf{x}, \mathbf{y}^t) - \nabla_{\mathbf{y}} g\left(\mathbf{x}, \mathbf{y}^{\mathbf{t}'}\right)\right) \end{bmatrix}$$
$$+ \alpha_x^2 \left\| \begin{bmatrix} \left(\nabla f(\mathbf{x}, \mathbf{y}) - \nabla f\left(\mathbf{x}', \mathbf{y}\right)\right) \\ \sum_{t=0}^{T-1} \left(\nabla_{\mathbf{y}} g(\mathbf{x}, \mathbf{y}^t) - \nabla_{\mathbf{y}} g\left(\mathbf{x}, \mathbf{y}^{\mathbf{t}'}\right)\right) \end{bmatrix} \right\|^2$$
$$= (1 - (\alpha_x \mu_f + \alpha_x \mu_g))^2 \left\| \begin{bmatrix} \mathbf{x} - \mathbf{x}' \\ \mathbf{y} - \mathbf{y} \end{bmatrix} \right\|^2 + \alpha_x^2 \left\| \begin{bmatrix} \left(\nabla \widetilde{f}(\mathbf{x}, \mathbf{y}) - \nabla \widetilde{f}\left(\mathbf{x}', \mathbf{y}\right)\right) \\ \sum_{t=0}^{T-1} \left(\nabla_{\mathbf{y}} \widetilde{g}(\mathbf{x}, \mathbf{y}^t) - \nabla_{\mathbf{y}} \widetilde{g}\left(\mathbf{x}, \mathbf{y}^{\mathbf{t}'}\right)\right) \end{bmatrix} \right\|^2$$
$$- 2\left(1 - \alpha_x \mu_f - \alpha_x \mu_g\right)\alpha_x \begin{bmatrix} \mathbf{x} - \mathbf{x}' \\ \mathbf{y} - \mathbf{y} \end{bmatrix}^T \begin{bmatrix} \left(\nabla \widetilde{f}(\mathbf{x}, \mathbf{y}) - \nabla \widetilde{f}\left(\mathbf{x}', \mathbf{y}\right)\right) \\ \sum_{t=0}^{T-1} \left(\nabla_{\mathbf{y}} \widetilde{g}(\mathbf{x}, \mathbf{y}^t) - \nabla_{\mathbf{y}} \widetilde{g}\left(\mathbf{x}, \mathbf{y}^{\mathbf{t}'}\right)\right) \end{bmatrix}$$
$$\leq \left(1 - 2\alpha \left(\mu_f + \mu_g\right) + \alpha^2 l^2\right) \|\mathbf{x} - \mathbf{x}'\|^2.$$

The penultimate inequality arises from the smoothness of $\widetilde{f}, \widetilde{g}$, which is based on our assumption for simplicity that $l = \max\{l_f, \frac{1+(\alpha_y l_g)^2}{(1-\alpha_y l_g)\alpha_y}\}$, and the details will be revealed as follows:

$$
l^2 \left\| \begin{bmatrix} \mathbf{x} - \mathbf{x}' \\ \mathbf{y} - \mathbf{y} \end{bmatrix} \right\|^2 \geq \left\| \begin{bmatrix} \nabla f(\mathbf{x}, \mathbf{y}) - \nabla f(\mathbf{x}', \mathbf{y}) \\ \sum_{t=0}^{T-1} \left( \nabla_{\mathbf{y}} g(\mathbf{x}, \mathbf{y}^t) - \nabla_{\mathbf{y}} g\left(\mathbf{x}, \mathbf{y}^{\mathbf{t}'}\right) \right) \end{bmatrix} \right\|^2
$$

$$
= \left\| \begin{bmatrix} \left( \nabla \widetilde{f}(\mathbf{x}, \mathbf{y}) - \nabla \widetilde{f}(\mathbf{x}', \mathbf{y}) \right) \\ \sum_{t=0}^{T-1} \left( \nabla_{\mathbf{y}} \widetilde{g}(\mathbf{x}, \mathbf{y}^t) - \nabla_{\mathbf{y}} \widetilde{g}\left(\mathbf{x}, \mathbf{y}^{\mathbf{t}'}\right) \right) \end{bmatrix} \right\|^2 + (\mu_f + \mu_g)^2 \left\| \begin{bmatrix} \mathbf{x} - \mathbf{x}' \\ \mathbf{y} - \mathbf{y} \end{bmatrix} \right\|^2
$$

$$
+ 2(\mu_f + \mu_g) \begin{bmatrix} \mathbf{x} - \mathbf{x}' \\ \mathbf{y} - \mathbf{y} \end{bmatrix}^T \begin{bmatrix} \left( \nabla \widetilde{f}(\mathbf{x}, \mathbf{y}) - \nabla \widetilde{f}(\mathbf{x}', \mathbf{y}) \right) \\ \sum_{t=0}^{T-1} \left( \nabla_{\mathbf{y}} \widetilde{g}(\mathbf{x}, \mathbf{y}^t) - \nabla_{\mathbf{y}} \widetilde{g}\left(\mathbf{x}, \mathbf{y}^{\mathbf{t}'}\right) \right) \end{bmatrix}
$$

$$
\geq \left\| \begin{bmatrix} \left( \nabla \widetilde{f}(\mathbf{x}, \mathbf{y}) - \nabla \widetilde{f}(\mathbf{x}', \mathbf{y}) \right) \\ \sum_{t=0}^{T-1} \left( \nabla_{\mathbf{y}} \widetilde{g}(\mathbf{x}, \mathbf{y}^t) - \nabla_{\mathbf{y}} \widetilde{g}\left(\mathbf{x}, \mathbf{y}^{\mathbf{t}'}\right) \right) \end{bmatrix} \right\|^2 + (\mu_f + \mu_g)^2 \left\| \begin{bmatrix} \mathbf{x} - \mathbf{x}' \\ \mathbf{y} - \mathbf{y} \end{bmatrix} \right\|^2 .
$$

Similar to the convex case, we can have:

$$
\left\| G_s \left( \begin{bmatrix} \mathbf{x} \\ \mathbf{y} \end{bmatrix} \right) - G_s \left( \begin{bmatrix} \mathbf{x}' \\ \mathbf{y}' \end{bmatrix} \right) \right\| \leq (1 + \alpha l) \left\| \begin{bmatrix} \mathbf{x} - \mathbf{x}' \\ \mathbf{y} - \mathbf{y}' \end{bmatrix} \right\| .
$$

$\square$

*Proof.* Because the main proof of Lemma 12 is similar to that of Lemma 9, we omit it. $\square$

Next, we give a bound for the update rule $G_T$ and prepare to prove Theorem 3.
Since $g()$ is a $l_g$-smooth function, we have:

$$
g\left(\mathbf{x}, \mathbf{y}^{t+1}\right) \leq g\left(\mathbf{x}, \mathbf{y}^t\right) + \left\langle \nabla g\left(\mathbf{x}, \mathbf{y}^t\right), \mathbf{y}^{t+1} - \mathbf{y}^t \right\rangle + \frac{l_g}{2} \left\| \mathbf{y}^{t+1} - \mathbf{y}^t \right\|^2 .
$$

$$
\leq g\left(\mathbf{x}, \mathbf{y}^t\right) - \left\langle \nabla g\left(\mathbf{x}, \mathbf{y}^t\right), \alpha_y \nabla g\left(\mathbf{x}, \mathbf{y}^t\right) \right\rangle + \frac{l_g}{2} \left\| \alpha_y \nabla g\left(\mathbf{x}, \mathbf{y}^t\right) \right\|^2
$$

$$
\leq g\left(\mathbf{x}, \mathbf{y}^t\right) - \alpha_y \left( 1 - \frac{\alpha_y l_g}{2} \right) \left\| \nabla g\left(\mathbf{x}, \mathbf{y}^t\right) \right\|^2 .
$$

The two sides are accumulated from $t = 1$ to $t = T$ and we could derive the following by *Cauchy–Schwarz* inequality:

$$
\sum_{t=1}^{T} \left\| \nabla g\left(\mathbf{x}, \mathbf{y}^t\right) \right\|^2 \leq \frac{g\left(\mathbf{x}, \mathbf{y}_1\right) - g\left(\mathbf{x}, \mathbf{y}_T\right)}{\alpha_y \left(2 - \alpha_y l_g\right)} \Rightarrow \left( \sum_{i=1}^{T} \nabla g(\mathbf{x}, \mathbf{y}^t) \right)^2 \leq T \sum_{i=1}^{T} \nabla g^2(\mathbf{x}, \mathbf{y}^t)
$$

$$
\leq \frac{T\left(g\left(\mathbf{x}, \mathbf{y}_1\right) - g\left(\mathbf{x}, \mathbf{y}_T\right)\right)}{\alpha_y(2 - \alpha_y l_g)} .
$$

Hence, the bound of $G_T$ equals to $\sqrt{L_f^2 \alpha_x^2 + \left( \frac{T(g(\mathbf{x}, \mathbf{y}_1) - g(\mathbf{x}, \mathbf{y}_T))}{\alpha_y(2 - \alpha_y l_g)} \right)^2}$. Now, we are ready to give the proof of Theorem 3.

*Proof of Part(a).* Suppose that $D_{m_1}$ and $D'_{m_1}$ are two neighboring sets differing in only one sample. Consider the updates $G_T^1, ..., G_T^K$ and $(G_T^1)', ..., (G_T^K)'$. We can observe that the example chosen by algorithm is the same in $D_{m_1}$, $D'_{m_1}$ at step $k$ with probability $1 - 1/m_1$ and different with probability $1/m_1$. Similarly to the previous single-timescale methods, in the former case, we have identical update rules, while $(1 + \alpha l)$-expansive can be employed in the latter through lemma 10.

$$\mathbb{E}\left[\delta_{k+1}\right] \leq \left(1 - \frac{1}{m_1}\right)(1 + \alpha l)\mathbb{E}\left[\delta_k\right] + \frac{1}{m_1}\mathbb{E}\left[\delta_k\right]$$

$$+ \frac{2}{m_1}\sqrt{L_f^2\alpha_x^2 + \left(\frac{T\left(g\left(\mathbf{x}, \mathbf{y}_1\right) - g\left(\mathbf{x}, \mathbf{y}_T\right)\right)}{\alpha_x(2 - \alpha_x l_g)}\right)^2}$$

$$= (1 + \alpha l)\mathbb{E}\left[\delta_k\right] + \frac{2}{m_1}\sqrt{L_f^2\alpha_x^2 + \left(\frac{T\left(g\left(\mathbf{x}, \mathbf{y}_1\right) - g\left(\mathbf{x}, \mathbf{y}_T\right)\right)}{\alpha_x(2 - \alpha_x l_g)}\right)^2}$$

$$\mathbb{E}\left[\delta_k\right] \leq \frac{2}{m_1}\sqrt{L_f^2\alpha_x^2 + \left(\frac{T\left(g\left(\mathbf{x}, \mathbf{y}_1\right) - g\left(\mathbf{x}, \mathbf{y}_T\right)\right)}{\alpha_x(2 - \alpha_x l_g)}\right)^2} \cdot \frac{(1 + \alpha l)^k - 1}{1 - 1 + \alpha}$$

$$\mathbb{E}\left[\delta_k\right] \leq \mathcal{O}\left(\frac{\sqrt{L_f^2\alpha_x^2 + (\frac{2T}{\alpha_y(2 - \alpha_y l_g)})^2}(1 + \alpha l)^k}{m_1}\right).$$

$\square$

The proof of Part(b) is the same as Part(a) and Part(c) are analogous to their counterparts in the single-timescale case. Thus, we omit them here.

### C.3.2 PARTICULAR SETTING

We introduce the following lemma as an extension of expansivity for the particular NC-SC setting.

**Lemma 13.** *Consider Problem (1) in the NC-SC setting and assume that Assumptions 1 and 2 hold. Suppose that $(\mathbf{x}, \mathbf{y})$ and $(\mathbf{x}', \mathbf{y}')$ are produced by Algorithm 2 with step size $\alpha_x = \alpha_y$. Let $\alpha l = \max\{\alpha_x l_f, \alpha_y l_g\}$. Then, we have the following expansivity equality:*

$$\left\|G_T\left(\begin{bmatrix}\mathbf{x}\\\mathbf{y}\end{bmatrix}\right) - G_T\left(\begin{bmatrix}\mathbf{x}'\\\mathbf{y}'\end{bmatrix}\right)\right\| \leq \begin{bmatrix}1 + \alpha_x lT & 0\\0 & 1 - \alpha_x\mu_g + \alpha_x lT\end{bmatrix}\begin{bmatrix}\|\ \mathbf{x} - \mathbf{x}'\ \|\\\|\ \mathbf{y} - \mathbf{y}'\ \|\end{bmatrix}.$$

*Proof.* By the triangle inequality, we have:

$$\left\|G_T\left(\begin{bmatrix}\mathbf{x}\\\mathbf{y}\end{bmatrix}\right) - G_T\left(\begin{bmatrix}\mathbf{x}'\\\mathbf{y}'\end{bmatrix}\right)\right\| \leq \left\|G_T\left(\begin{bmatrix}\mathbf{x}\\\mathbf{y}\end{bmatrix}\right) - G_T\left(\begin{bmatrix}\mathbf{x}'\\\mathbf{y}\end{bmatrix}\right)\right\| + \left\|G_T\left(\begin{bmatrix}\mathbf{x}'\\\mathbf{y}\end{bmatrix}\right) - G_T\left(\begin{bmatrix}\mathbf{x}'\\\mathbf{y}'\end{bmatrix}\right)\right\|.$$

The first item can be derived from:

$$\left\|G_T\left(\begin{bmatrix}\mathbf{x}\\\mathbf{y}\end{bmatrix}\right) - G_T\left(\begin{bmatrix}\mathbf{x}'\\\mathbf{y}\end{bmatrix}\right)\right\| = \left\|\begin{bmatrix}\mathbf{x} - \mathbf{x}' - \alpha_x\left(\nabla f(\mathbf{x}, \mathbf{y}) - \nabla f\left(\mathbf{x}', \mathbf{y}\right)\right)\\\mathbf{y} - \mathbf{y} + \alpha_y \sum_{y=1}^T\left(\nabla_\mathbf{y} g(\mathbf{x}, \mathbf{y}^t) - \nabla_\mathbf{y} g\left(\mathbf{x}', \mathbf{y}^t\right)\right)\end{bmatrix}\right\|$$

$$\leq (1 + \alpha Tl)\left\|\mathbf{x} - \mathbf{x}'\right\|.$$

If $g$ is strongly-convex, then $\widetilde{g}(\mathbf{x}, \mathbf{y}) = g(\mathbf{x}, \mathbf{y}) - \frac{\mu_g}{2}(\|x\|^2 + \|y\|^2)$ will be convex. With the monotonicity of $\widetilde{g}$ gradient,, we can derive the following:

$$\left\|G_T\left(\begin{bmatrix}\mathbf{x}'\\\mathbf{y}\end{bmatrix}\right) - G_T\left(\begin{bmatrix}\mathbf{x}'\\\mathbf{y}'\end{bmatrix}\right)\right\|^2$$

$$= \left\|\begin{bmatrix}\mathbf{x}' - \mathbf{x}'\\\mathbf{y} - \mathbf{y}'\end{bmatrix}\right\|^2 - 2\alpha_x\begin{bmatrix}\mathbf{x}' - \mathbf{x}'\\\mathbf{y} - \mathbf{y}'\end{bmatrix}^T\begin{bmatrix}(\nabla f(\mathbf{x}', \mathbf{y}) - \nabla f(\mathbf{x}', \mathbf{y}'))\\(\nabla_\mathbf{y} g(\mathbf{x}', \mathbf{y}) - \nabla_\mathbf{y} g(\mathbf{x}', \mathbf{y}'))\end{bmatrix}$$

$$+ \alpha_x^2\left\|\begin{bmatrix}(\nabla f(\mathbf{x}', \mathbf{y}) - \nabla f(\mathbf{x}', \mathbf{y}'))\\(\nabla_\mathbf{y} g(\mathbf{x}', \mathbf{y}) - \nabla_\mathbf{y} g(\mathbf{x}', \mathbf{y}'))\end{bmatrix}\right\|^2$$

$$= (1 - \alpha_x\mu_g)^2\left\|\begin{bmatrix}\mathbf{x}' - \mathbf{x}'\\\mathbf{y} - \mathbf{y}'\end{bmatrix}\right\|^2 + \alpha_x^2\left\|\begin{bmatrix}(\nabla f(\mathbf{x}', \mathbf{y}) - \nabla f(\mathbf{x}', \mathbf{y}'))\\(\nabla_\mathbf{y} \widetilde{g}(\mathbf{x}', \mathbf{y}) - \nabla_\mathbf{y} \widetilde{g}(\mathbf{x}', \mathbf{y}'))\end{bmatrix}\right\|^2$$

$$- 2(1 - \alpha_x\mu_g)\alpha_x\begin{bmatrix}\mathbf{x}' - \mathbf{x}'\\\mathbf{y} - \mathbf{y}'\end{bmatrix}^T\begin{bmatrix}(\nabla f(\mathbf{x}', \mathbf{y}) - \nabla f(\mathbf{x}', \mathbf{y}'))\\(\nabla_\mathbf{y} \widetilde{g}(\mathbf{x}', \mathbf{y}) - \nabla_\mathbf{y} \widetilde{g}(\mathbf{x}', \mathbf{y}'))\end{bmatrix}$$

$$\leq ((1 - \alpha_x\mu_g)^2 + \alpha_x^2 l^2 T^2)\|\mathbf{y} - \mathbf{y}'\|^2.$$

Hence, the second item can be derived $\left\| G_T \left( \begin{bmatrix} \mathbf{x}' \\ \mathbf{y} \end{bmatrix} \right) - G_T \left( \begin{bmatrix} \mathbf{x}' \\ \mathbf{y}' \end{bmatrix} \right) \right\| \leq (1 - \alpha_x \mu_g + \alpha_x l T) \| \mathbf{y} - \mathbf{y}' \|$. $\hfill\square$

Next, we consider the extension of the growth lemma:

**Lemma 14.** *Consider two sequences of updates $G_T^1, ..., G_T^K$ and $(G_T^1)', ..., (G_T^K)'$ with initial points $\mathbf{x}_0 = \mathbf{x}_0'$, $\mathbf{y}_0 = \mathbf{y}_0'$. Define $\delta_{\mathbf{x},k} = \| \mathbf{x}_k - \mathbf{x}_k' \|$ and $\delta_{\mathbf{y},k} = \| \mathbf{y}_k - \mathbf{y}_k' \|$. Suppose that $G_T^k$ is $\eta$-expansive and for every $(\mathbf{x}_{G_T^k}, \mathbf{y}_{G_T^k}) := G_T^k(\mathbf{x}, \mathbf{y})$, $(\mathbf{x}_{(G_T^k)'}, \mathbf{y}_{(G_T^k)'}) := (G_T^k)'(\mathbf{x}, \mathbf{y})$ and*

$$\sup_{\mathbf{x},\mathbf{y}} \left\| \mathbf{x}_{G_T^k} - \mathbf{x} \right\| \leq \sigma_x, \qquad \sup_{\mathbf{x},\mathbf{y}} \left\| \mathbf{y}_{G_T^k} - \mathbf{y} \right\| \leq \sigma_y,$$
$$\sup_{\mathbf{x},\mathbf{y}} \left\| \mathbf{x}_{(G_T^k)'} - \mathbf{x} \right\| \leq \sigma_x, \quad \sup_{\mathbf{x},\mathbf{y}} \left\| \mathbf{y}_{(G_T^k)'} - \mathbf{y} \right\| \leq \sigma_y.$$

*Then, we have*

$$\begin{bmatrix} \delta_{x,k+1} \\ \delta_{y,k+1} \end{bmatrix} \leq \eta \begin{bmatrix} \delta_{x,k} \\ \delta_{y,k} \end{bmatrix} + 2 \begin{bmatrix} \sigma_x \\ \sigma_y \end{bmatrix}.$$

Now, we are ready to give the proof of Theorem 4.

*Proof.*

$$\begin{bmatrix} \mathbb{E}\left[ \delta_{x,k+1} \right] \\ \mathbb{E}\left[ \delta_{y,k+1} \right] \end{bmatrix} \leq \left( 1 - \frac{1}{m_1} \right) \begin{bmatrix} 1 + \frac{cl}{k} & 0 \\ 0 & 1 + \frac{c}{k}(Tl - \mu_g) \end{bmatrix}$$
$$+ \frac{1}{m_1} \left( \begin{bmatrix} 1 + \frac{cl}{k} & 0 \\ 0 & 1 + \frac{c}{k}(Tl - \mu_g) \end{bmatrix} \begin{bmatrix} \mathbb{E}\left[ \delta_{x,k} \right] \\ \mathbb{E}\left[ \delta_{y,k} \right] \end{bmatrix} + \begin{bmatrix} \frac{2cl_f}{k} \\ \frac{2cl_g T}{k} \end{bmatrix} \right)$$
$$\leq \begin{bmatrix} 1 + \frac{cl}{k} & 0 \\ 0 & 1 + \frac{c}{k}(Tl - \mu_g) \end{bmatrix} \begin{bmatrix} \mathbb{E}\left[ \delta_{x,k} \right] \\ \mathbb{E}\left[ \delta_{y,k} \right] \end{bmatrix} + \begin{bmatrix} \frac{2cl_f}{m_1 k} \\ \frac{2cl_g T}{m_1 k} \end{bmatrix}$$
$$\leq \begin{bmatrix} \mathbb{E}\left[ \delta_{x,K} \right] \\ \mathbb{E}\left[ \delta_{y,K} \right] \end{bmatrix} \leq t \sum_{k=k_0+1}^{K} \left\{ \prod_{t=k+1}^{K} \begin{bmatrix} 1 + \frac{cl}{k} & 0 \\ 0 & 1 + \frac{c}{k}(Tl - \mu_g) \end{bmatrix} \right\} \begin{bmatrix} \frac{2cl_f}{m_1 k} \\ \frac{2cl_g T}{m_1 k} \end{bmatrix}$$
$$\leq \sum_{k=k_0+1}^{K} \begin{bmatrix} \exp\left( \sum_{t=k+1}^{K} \frac{cl}{k} \right) & 0 \\ 0 & \exp\left( \sum_{t=k+1}^{K} \frac{c}{k}(Tl - \mu_g) \right) \end{bmatrix} \begin{bmatrix} \frac{2cl_f}{m_1 k} \\ \frac{2cl_g T}{m_1 k} \end{bmatrix}$$
$$\leq \frac{2c\sqrt{l_f^2 + T^2 l^2}}{m_1} \sum_{k=k_0+1}^{K} \frac{\exp\left( \sum_{t=k+1}^{k} \frac{c}{k}(Tl + l - \mu_g) \right)}{k}$$
$$\leq \frac{2c\sqrt{l_f^2 + T^2 l_g^2} K^{-c(Tl+l-\mu_g)}}{m_1} \sum_{k=k_0+1}^{K} k^{-c(Tl+l-\mu_g)-1}$$
$$\leq \frac{2\sqrt{l_f^2 + T^2 l_g^2}}{m_1(Tl + l - u_g)} \cdot \left( \frac{K}{k_0} \right)^{c \cdot (Tl+l-\mu_g)}.$$

According to Lemma 11, we have:

$$\mathbb{E}\left[ |f(\mathbf{x}, \mathbf{y}) - f(\mathbf{x}', \mathbf{y}')| \right] \leq \frac{k_0}{m_1} + \frac{2L_f \sqrt{l_f^2 + T^2 l_g^2}}{m_1(Tl + l - \mu_g)} \left( \frac{K}{k_0} \right)^{c(Tl+l-\mu_g)}.$$

Let $p = Tl + l - \mu_g$. The right hand side is approximately minimized when

$$k_0 = \left( 2cL_f \sqrt{l_f^2 + T^2 l_g^2} \right)^{\frac{1}{cp+1}} \cdot K^{\frac{cp}{cp+1}}.$$

Therefore, we have

$$\beta \leq \mathcal{O}\left(\frac{\left(2cL_f\sqrt{l_f^2 + l_g^2 T^2}\right)^{\frac{1}{cp+1}} \cdot K^{\frac{cp}{cp+1}}(p + 2/c)}{m_1 p}\right).$$

$\square$

### C.4 THE PROOF OF COROLLARY

*Proof of Corollary 5.* Based on the proof of previous result in C.2, we have:

$$\mathbb{E}\left[|f(x, y) - f(x', y')|\right] \leq \frac{k_0}{m_1} + \frac{2L_f\sqrt{l_f^2 + l_g^2}}{m_1 l}\left(\frac{K}{k_0}\right)^{cl}.$$

The right hand side is approximately minimized when

$$k_0 = \left(2cL_f\sqrt{l_f^2 + l_g^2}\right)^{\frac{1}{cl+1}} \cdot K^{\frac{cl}{cl+1}}.$$

Therefore, we have

$$\mathbb{E}\left[|f(x, y) - f(x', y')|\right] \leq \mathcal{O}\left(\frac{\left(2cL_f\sqrt{l_f^2 + l_g^2}\right)^{\frac{1}{cl+1}} \cdot K^{\frac{cl}{cl+1}}(l + 2/c)}{m_1 l}\right)$$

$$= \mathcal{O}(K^{\frac{cl}{cl+1}}/m_1).$$

$\square$

*Proof of Corollary 6.* Based on the previous result, we have:

$$\mathbb{E}\left[|f(x, y) - f(x', y')|\right] \leq \frac{k_0}{m_1} + \frac{2L_f\sqrt{l_f^2 + T^2 l_g^2}}{m_1 T l}\left(\frac{K}{k_0}\right)^{Tcl}.$$

The right hand side is approximately minimized when

$$k_0 = \left(2cL_f\sqrt{l_f^2 + l_g^2}\right)^{\frac{1}{Tcl+1}} \cdot K^{\frac{Tcl}{Tcl+1}}.$$

Therefore, we have

$$\mathbb{E}\left[|f(x, y) - f(x', y')|\right] \leq \mathcal{O}\left(\frac{\left(2cL_f\sqrt{l_f^2 + T^2 l_g^2}\right)^{\frac{1}{Tcl+1}} \cdot K^{\frac{Tcl}{Tcl+1}}(Tl + c/2)}{m_1 T l}\right)$$

$$= \mathcal{O}\left(\frac{T^{\frac{1}{Tcl+1}} K^{1 - \frac{1}{Tcl+1}}}{m_1}\right).$$

$\square$

# D ADDITIONAL EXPERIMENT

## D.1 META LEARNING

We also conduct other experiment with single timescale (Algorithm 1) on meta learning to validate our theoretical findings. The model overfits with large value of $K$, which matches our theorem 2.

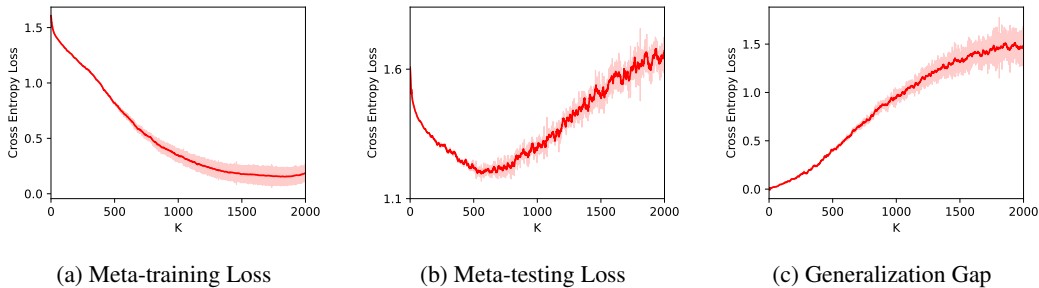

(a) Meta-training Loss                (b) Meta-testing Loss                (c) Generalization Gap

Figure 2: Results of Meta Learning with single timescale optimization

An additional experiment on the MNIST dataset is conducted for Theorem 4, following the same settings on Omnilot dataset. Both Figure 1c and Figure 3c validate the influence of $K$ and $T$ on the generalization gap. When $K$ is relatively small, $T$ is dominant since the gap with $T = 8$ is higher than that with $T = 2, 4$. When $K$ is large, The effect of $T$ fades and it contributes less to the trend of the gap.

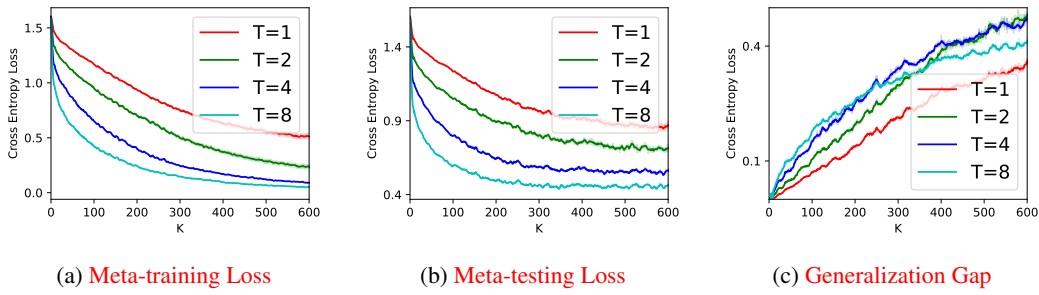

(a) Meta-training Loss                (b) Meta-testing Loss                (c) Generalization Gap

Figure 3: Results of Meta Learning with single timescale optimization

## D.2 HYPERPARAMETER OPTIMIZATION

Hyperparameter optimization (HO) is also an instance of bilevel optimization that minimizes the validation error of a model parameterized by $w$ with respect to hyperparameter $\lambda$. Mathematically, the objective function could be given by:

$$\min_\lambda \mathcal{L}_{\mathcal{D}_{\text{val}}}(\lambda) = \mathbb{E}_{\xi \in \mathcal{D}_{\text{val}}} \mathcal{L}(w^*; \xi), \tag{8a}$$

$$\text{s.t. } w^* = \arg\min_w \mathbb{E}_{\xi \in \mathcal{D}_{\text{tr}}} \left( \mathcal{L}(w, \lambda; \xi) + \mathcal{R}_{w, \lambda} \right), \tag{8b}$$

where $\mathcal{D}_{\text{tr}}$ and $\mathcal{D}_{\text{val}}$ are the training and validation sets and $\mathcal{R}_{w, \lambda}$ is the regularizer. In the inner level, the procedure optimizes $w$ using the training set (Equation 8b). In the outer level, it optimizes $\lambda$ using the validation set (Equation 8a).

**Settings and Implementation** We adopt the task of data hyper-cleaning. It aims to reweight data samples with the noisy label. Therefore, the hyperparameter $\lambda$ is the weight of each sample in the training set. We follow a similar setting in Bao et al. (2021) on the MNIST dataset (LeCun, 1998), which consists of greyscale hand-written digits with size $28 \times 28$. The model $w$ corresponds to the classification network and the hyperparameter $\lambda$ corresponds to the weights of each individual training sample. We establish the experiment using PyTorch (Paszke et al., 2019). The model $w$ is

a 2-layer fully connected network with size $784 \rightarrow 256 \rightarrow 10$ for the 10 digit classification. The hyperparameter $\lambda$ is a weighting vector with length 2000 for all training samples. We randomly sample 2000, 1000 and 1000 images for the training, validation and testing set. Training samples are corrupted with the probability of 50%, *i.e.,* roughly half samples are labeled with random and wrong values, instead of the correct ones. We train $w$ using the training set in the inner level and $\lambda$ using the validation set in the outer level. The batch size is 32 and the learning rate for $w$ and $\lambda$ are 0.01 and 10, respectively. Results are evaluated based on the average of 5 trial runs with different random seeds.

**Results Evaluation** Figure 4 demonstrates the results of Algorithm 2 on the regularized HO problem where the inner level function is often strongly-convex. It is clearly shown in Figure 4b that $T = 8$ causes the increase of testing loss and indicates that the risk of model overfitting increases as $T$ rises. Additionally, the generalization gap maintains a consistent relationship with both inner and outer iterations, $T$ and $K$ respectively, which is corresponding to our Theorem 4.

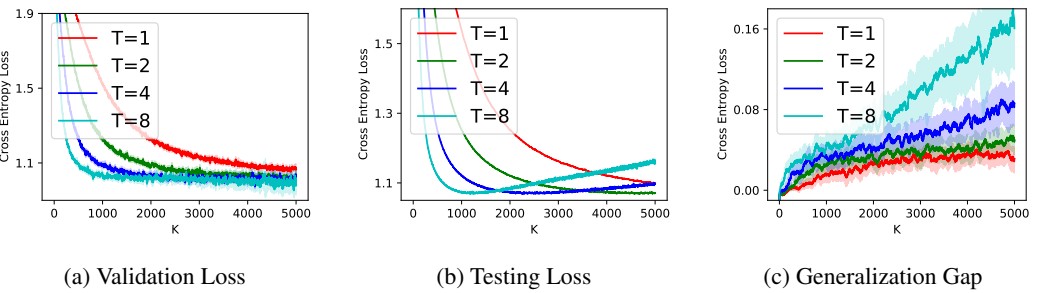

(a) Validation Loss  (b) Testing Loss  (c) Generalization Gap

Figure 4: Results of hyperparameter optimization with various values of $T$ and $K$

We also conduct the experiment with smaller learning rate 0.001 and larger number of steps $\{64, 128, 256\}$ in the inner level. Compared to Figure 4, Figure 5 presents similar behavior in terms of the effect of the value of $K$ and $T$, while Figure 5c performs a higher variance caused by the accumulation of inner level updates. Furthermore, we can show that performance on the test dataset is comparable to performance with smaller inner iteration even when inner iteration $T$ substantially increases, suggesting the effectiveness of TSGD and validating our Theorem 4.

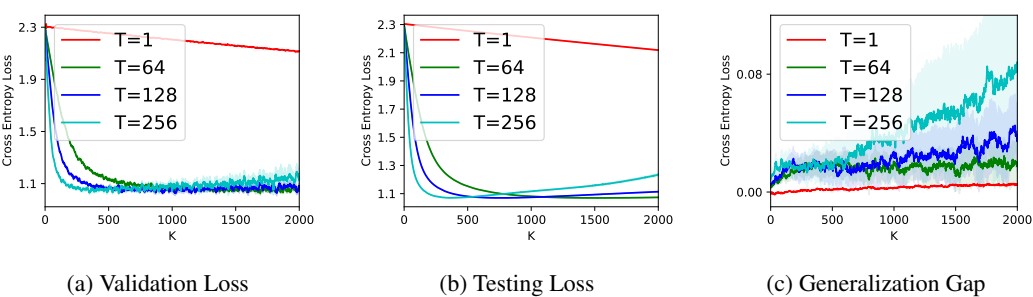

(a) Validation Loss  (b) Testing Loss  (c) Generalization Gap

Figure 5: Results of hyperparameter optimization with large values of $T$

# E    EXISTING GAP IN THE ANALYSIS OF UD

The proofs in (Bao et al., 2021) make us suspect that $\theta(\lambda)$ is more or less treated as an argument independent of $\lambda$, even though in the description $\theta(\lambda)$ is said to be dependent on $\lambda$.

In the proof of Theorem 2 (Appendix A.2 of (Bao et al., 2021), page 14), the following equations are given,

$$\ell\left(\mathbf{A}\left(S^{tr}, S^{val}\right), z\right) = \ell\left(\lambda_t, \hat{\theta}\left(\lambda_t, S^{tr}\right), z\right) = f\left(\lambda_t, \hat{\theta}\right)$$

$$\ell\left(\mathbf{A}\left(S^{tr}, S'^{val}\right), z\right) = \ell\left(\lambda'_t, \hat{\theta}\left(\lambda'_t, S^{tr}\right), z\right) = f\left(\lambda'_t, \hat{\theta}\right).$$

This implies that $\hat{\theta}\left(\lambda_t, S^{tr}\right) = \hat{\theta}$ and $\hat{\theta}\left(\lambda'_t, S^{tr}\right) = \hat{\theta}$, which seems to suggest that $\hat{\theta}(\lambda)$ is independent of $\lambda$ which may conflict with the dependence on $\lambda$. Suppose (Bao et al., 2021) uses improper notations here, but in their following proof, there is still something confusing:

To understand the issue better, let $\delta_t = \|\lambda_t - \lambda'_t\|$. Suppose that $0 \leq t_0 \leq t$. They have the following inequality in the proof.

$$\mathbf{E}\left[\left|f\left(\lambda_t, \hat{\theta}\right) - f\left(\lambda'_t, \hat{\theta}\right)\right|\right] = \mathbf{E}\left[\left|f\left(\lambda_t, \hat{\theta}\right) - f\left(\lambda'_t, \hat{\theta}\right)\right| \cdot 1_{\delta_{t_0}=0}\right] + \mathbf{E}\left[\left|f\left(\lambda_t, \hat{\theta}\right) - f\left(\lambda'_t, \hat{\theta}\right)\right| \cdot 1_{\delta_{t_0}>0}\right]$$

$$\leq L\mathbf{E}\left[\delta_t \cdot 1_{\delta_{t_0}=0}\right] + P\left(\delta_{t_0} > 0\right)s(\ell).$$

The left hand side is to measure the expected difference of function $f$ with arguments $(\lambda_t, \hat{\theta})$. The first equation decomposes the left hand side according to the two possible cases of $\delta_{t_0}$ (i.e., $\delta_{t_0} = 0$ or $\delta_{t_0} > 0$). The first term of the inequality is derived from the Lipschitz continuous property of $f$. However, to use the Lipschitz continuity of the multivariate function $f$ with respect to $\hat{\theta}$, i.e.,

$$|f(\lambda_t, \hat{\theta}) - f(\lambda'_t, \hat{\theta})| \leq L\|\lambda_t - \lambda'_t\|,$$

$\hat{\theta}$ needs to be the same varible (i.e., they have the same value all the time) in both $f(\lambda_t, \hat{\theta})$ and $f(\lambda'_t, \hat{\theta})$. However, from the UD algorithm in (Bao et al., 2021) it is clear that $\hat{\theta}$ will not always have the same value in $f(\lambda_t, \hat{\theta})$ and $f(\lambda'_t, \hat{\theta})$ when $\lambda$ changes.

Specifically, when $t = t_0, \delta_{t_0} = 0, \lambda_{t_0} = \lambda'_{t_0}$, we have $\hat{\theta}_K^{t_0+1}(\lambda_{t_0}, S^{tr}) = \hat{\theta}'^{t_0+1}_K(\lambda'_{t_0}, S^{tr})$. Since $S^{val}$ and $S'^{val}$ are assumed to differ by at most one point, without loss of generality, we suppose that SGD selects the different point at timestep $t_s$. Then, we have

$$t = t_s$$
$$\lambda_{t_s}(\lambda_{t_s-1}, \hat{\theta}_{t_s-1}, S^{val}) \neq \lambda'_{t_s}(\lambda_{t_s-1}, \hat{\theta}_{t_s-1}, S'^{val})$$
$$\delta_{t_s} \neq 0$$
$$\hat{\theta}_K^{t_s+1}(\lambda_{t_s}, S^{tr}) \neq \hat{\theta}'^{t_s+1}_K(\lambda'_{t_s}, S^{tr}).$$

This means that $\hat{\theta}_K^{t_s+1}(\lambda_{t_s}, S^{tr}) \neq \hat{\theta}'^{t_s+1}_K(\lambda'_{t_s}, S^{tr})$ for all $t \geq t_s$, as the update of $\hat{\theta}$ and $\lambda$ use the value of $\hat{\theta}$ in the previous iteration. Thus, we cannot use the Lipschitz property to derive the first term in the aforementioned inequality. That is why we think there may exist some gap in the analysis of UD algorithms in (Bao et al., 2021).

