# OpenReview forum: "On Stability and Generalization of Bilevel Optimization Problems"
_ICLR.cc/2023/Conference — Submitted to ICLR 2023_

### Official Review · Reviewer_Wyf6 · 2022-10-23

**Confidence:** 5
**Correctness:** 2
**Technical Novelty And Significance:** 2
**Empirical Novelty And Significance:** 2
**Recommendation:** 1

**Clarity, Quality, Novelty And Reproducibility:**

There are some typos in paper and should be double checked. The technique in this work is relatively less novel, since its technique mainly follows [1,2].


**Strength And Weaknesses:**

**Strength**

This paper makes the following extensions compared to [1]:

1. This paper considers the stability bound of SSGD and TSGD algorithms, which is not studied in [1]

2. This paper also studies the case where the outer level problem is convex or strongly convex.

3. This paper provides a better high probability bound from $O(\beta \sqrt{m_1})$ to $O(\beta \log m_1)$ compared to [1]. The improvement is significant.



**Weakness**

There are some improper or incorrect claims on the prior work [1], upon which this work is built:

1. [1] studies the UD algorithm, and it seems that the author has some misunderstanding on the UD algorithm. Indeed, UD view $y$ produced by the inner level optimization as a function of $x$, i.e., $y(x) = H_{T-1} \circ H_{T-2} \circ \cdots \circ H_{0} (x)$, where $H_t$ represents the gradient update in Line 6 of Algorithm 3. When UD optimizes $x$, it would use the gradient $\nabla_x f(x, y(x); D_{m_1}) = \nabla_x f(x, y; D_{m_1}) + \nabla_x y(x) \nabla_y f(x, y; D_{m_1})$, instead of $\nabla_x f(x, y; D_{m_1})$ in Line 8 of Algorithm 3. This means that UD would backward through the optimization trajectory of $y$. However, as shown in Remark 4, the author thinks UD treat $y(x)$ as an argument indpendent $x$, which is an incorrect claim. As a result, the claim "However, there are some technical flaws
in their analysis..." in Section 1, and the claim "...and thus cannot be treated as an argument independent of $\lambda_t'$, which is misused and causes technical flaws in their proof subsequently" in Remark 4 are both incorrect.

2. Since this work consider totally two different algorithms SSGD and TSGD other than UD. Perhaps it is less proper to state "for the SC-SC and C-C cases with single-timescale update strategy we significantly improve the generalization bounds compared with [1]", since the algorithm already changes.

3. This work claims that "[1] only considers a general setting for inner function" in Section 1. However, [1] also considers convex and strongly convex inner functions. These results are provided in Appendix C in [1].

4. This work claims that the bound of [1] "is quite loose in some cases" in Section 1. However, [1] constructs a worst case (see Appendix B in [1]) to prove that the bound of [1] is tight if no extra assumptions of inner or outer functions are provides. Besides, [1] also gives tighter bound when convex or strongly convex assumptions are made. Therefore, the claim that the bound of [1] "is quite loose in some cases" is improper.

5. This work claims that [1] has an undesirable issue "the stability for general bilevel optimization is still unknown". However, the problem formulation of bilevel optimization (see Eq.(2)) in this work is exactly the same as [1]. Therefore, the range of bilevel optimization considered in this work is not more general than [1] in theory, and this undesirable issue remains in this work. As a result, the claim "our work is the first thorough generalization analysis for general bilevel optimization problem" is improper.

Other questions:

5. The notion of argument stability is a special case of uniform stability, and it is obvious that a $\beta$-argument-stable (in expectation) algorithm is also a $L_f \beta$-uniform-stable (in expectation) algorithm. Is it necessary to introduce this additional notion?

6. Theorem (1) c assumes the algorithm $A$ is uniform stable almost surely. This assumption looks quite strong for random algorithms, since it requires for all possible randomness in the algorithm, changing a data point in the validation set won't cause the loss to change more than $\beta$. Can SSGD and TSGD satisfy this assumption? It seems that the author does not verify this assumption for the studied SSGD and TSGD algorithms, and does not establish high probability bounds for the two algorithms.

7. In Talbe 1, the TSGD with C-C setting has a $O(T^K/m_1)$ bound, and TSGD with NC-NC setting has a $O(T^{1-\kappa_6} K^{\kappa_6} / m_1)$ bound. The former is much looser. Why a stronger assumption leads to a looser bound?

Typos:

8. In the second line below Eq.(2), $g(x, y(x); \xi_i)$ shoud be $g(x, y; \xi_i)$.

9. In the third line in the seoncd paragraph of Section 2.2, $f(x, y(x); \xi)$ should be $f(x, y; \xi)$.

10. In Line 7 of Algorithm 2, $x_k^t$ should be $x_k$.

[1] Stability and Generalization of Bilevel Programming in Hyperparameter Optimization

[2] Train faster, generalize better: Stability of stochastic gradient descent

**Summary Of The Paper:**

This work studies the generalization of bilevel optimization problem. Specifically, this work analyzes the SSGD and TSGD bilevel optimization algorithms. This work adopts the notion of uniform stability on validation in expectation [1], and analyze the stability coefficient $\beta$ to build a generalization bound of the two algorithms. This work also introduces the notion of argument stability (a special case of uniform stability), and provides a high probability bound for almost surely uniform stable algorithms.

**Summary Of The Review:**

It is nice to study the generalization of SSGD and TSGD bilevel optimization algorithms, since they are commonly used. However, many parts in this work need to be revised before being ready for publishment. For example, the author should carefully check the prior work [1], and revise claims on this work; the bound of TSGD in Table 1 should be checked and discussed; the writting should be double checked.

---

> ### Author Response · Authors · 2022-11-18
> **Response to the Reviewer Wyf6**
>
> We thank the reviewer for the detailed and constructive comments. Below please find our responses to your comments.
>
> - Please see details in our response to common concern 1.
>
> - This work and [1] analyze the generalization of bilevel optimization from different perspectives. [1] considers that different assumptions on functions affect the Lipschitz constants in Appendix C in [1], resulting in generalization boundary changes. Our work focuses on how the changes on the assumptions of functions could influence the stability directly with maintenance of Lipschitz constants. Towards remark (2, 4, 5), we revised our paper accordingly to address the raised issues.
>
> - The argument stability introduced by [2] measures the impact of changing a single training example on hypothesis, which is stronger than uniform stability and allows one to exploit martingale inequalities in the Banach space of the hypotheses.
>
> - One can still apply Theorem (1.c) if the almost sure stability is relaxed to uniform stability with high probability. While almost sure uniform stability is strong for randomized algorithms, several existing work show that randomized algorithms (e.g. SGD) are uniformly stable with high probability. Therefore, to apply Theorem (1.c) to SSGD and TSGD, it suffices to establish high-probability bounds for uniform stability of SSGD and TSGD, which we leave as future work.
>
> - We have recently improved our results for the strongly-convex-strongly-convex and convex-convex cases, which provide a novel perspective to estimate the stability bounds for the two-timescale algorithms. Please refer to Theorem 3 in the revised paper for details. Compared to the single-level problem in existing work [3], bilevel optimization is much more challenging due to the facts that it involves two functions and the algorithm TSGD has a two-layer loop, while the problem in [3] has only one function and its simple SGD algorithm has only one layer in the loop. Because of such difference, the inner loop in our TSGD will accumulate gradients from multiple SGD steps, which could make some nice properties such as convexity or strong convexity extremely difficulty to be exploited. Following the framework of algorithmic stability analysis in [3], we investigate the bilevel problem by employing the expansivity of update rules (please see details in Lemma 13). In the single level problem, e.g. [3], gradient update is non-expansive in the convex case and contractive in the strongly-convex case. In TSGD, the inner function has an accumulation of multiple SGD steps, which may have a negative impact on these properties. Actually this phenomenon has also occurred in recent work [4] for minmax problems. This seems to be an interesting problem for future research to further examine this phenomenon.
>
> - Originally, we used "general" to mean that our inner level parameter $\theta$ is dependent on $\lambda$, which is a more general case in practice (please see details in common concern 1). After comparing the definitions of the two problems, we agree with the reviewer and removed the "general" word from the paper. We thank the reviewer for pointing out this.
>
> - Thanks for pointing out the typos, we have fixed them in the revised paper.
>
> [1] Bao, Fan, et al. "Stability and Generalization of Bilevel Programming in Hyperparameter Optimization." Advances in Neural Information Processing Systems 34 (2021): 4529-4541.
>
> [2] Liu, Tongliang, et al. "Algorithmic stability and hypothesis complexity." International Conference on Machine Learning. PMLR, 2017.
>
> [3] Hardt, Moritz, Ben Recht, and Yoram Singer. "Train faster, generalize better: Stability of stochastic gradient descent." International conference on machine learning. PMLR, 2016.
>
> [4] Farnia, Farzan, and Asuman Ozdaglar. "Train simultaneously, generalize better: Stability of gradient-based minimax learners." International Conference on Machine Learning. PMLR, 2021.

---

### Official Review · Reviewer_jxJW · 2022-10-25

**Confidence:** 2
**Clarity, Quality, Novelty And Reproducibility:** Paper is rather clear and original
**Correctness:** 4
**Technical Novelty And Significance:** 4
**Empirical Novelty And Significance:** 3
**Recommendation:** 6

**Strength And Weaknesses:**

Major concerns:
- Even though authors provide a long list of points comparing their work against Bao et al 2021, I am not sure I understood the difference:
  - (1) "Their study is only for the hyper-parameter" I do not see how the class of problem you consider is more general
  - (4) Could you comment on why exactly their analysis needs a renitilization of the inner parameters?
  - (5) I agree, but you do not especially seem to provide extensive experimental results
Could you comment on this?

- **Experimental part** Except from the sentence "The trend of generalization error in terms of K and T matches with our analysis in Theorem 4", there are no links between the experiments and the proposed analysis.
Would it be possible to obtain more quantitative results than "a trend"?
I am not asking for SOTA experiments, but it would be nice to have quantitative experiments validating the provided theorems. For instance, would it be possible to provide experiments in the setting of Theorem 4? to compute the value of $\beta$, and check its variation as a function of T and K?


**Summary Of The Paper:**

Authors propose to study the stability of estimators based on bilevel optimization problems. Using the notion of algorithmic stability, authors managed to bound the generalization error. This analysis is proposed for usual bilevel optimization algorithms (single-timescale and two-timescale SGD).

**Summary Of The Review:**

Contribution seems interesting and new. I would appreciate if the distinction with previous work was clearer, and if experiments validate more the data

---

> ### Author Response · Authors · 2022-11-18
> **Response to the Reviewer jxJW**
>
> We thank the reviewer for the thoughtful comments and positive rating. Below are our responses to the raised concerns.
>
> - Originally, we used "general" to mean that our inner level parameter $\theta$ is dependent on $\lambda$, which is a more general case in practice (please see details in common concern 1). After comparing the definitions of the two problems, we agree with the reviewer and removed the "general" word from the paper. We thank the reviewer for pointing out this.
>
> - Please see line 4 in Algorithm 3. They take the initialization value before each entry.
>
> - Extremely high number of iterations (K for SSGD and K, T for TSGD) will drastically reduce the stability of these algorithms and increase the generalization error, which will make these algorithms prone to overfitting.
>
> - The experiment uses generalization gap (Figure 1c) to empirically validate the bound of Theorem 4. In Figure 1c, we provide the error curve in terms of values of $T$ and $K$, to show its trend of it. When $K$ is relatively small, $T$ is dominant since the gap with $T = 8$ is higher than that with $T = 2, 4$. When $K$ is large, The effect of $T$ fades and it contributes less to the trend of the gap. This match the formulation of our Theorem 4. To extensively validate it, we have added an additional experiment on the MNIST dataset which also shows similar trend. Please see the Appendix D.1 in our revised paper for more details.

---

### Official Review · Reviewer_GoE2 · 2022-10-26

**Confidence:** 4
**Correctness:** 2
**Technical Novelty And Significance:** 2
**Empirical Novelty And Significance:** Not applicable
**Recommendation:** 3

**Clarity, Quality, Novelty And Reproducibility:**

**Clarity**

The writing is clear. However, some of the claims are not well supported. See Weaknesses – Incorrectness. And Weaknesses-Lack of Clarity.


**Quality**
The paper deals with an important problem of algorithm stability in bilevel optimization. The paper and the proof is clearly written. Experiments on meta-learning and data reweighting are conducted.


**Novelty**
Though prior works exist that consider algorithm stability of bilevel learning, this paper considers two different algorithms without reinitialization in the inner loop.

**Strength And Weaknesses:**

**Strengths**

* The paper is clearly written and easy to follow.
* The paper considers two algorithms SSGD, and TSGD without reinitialization in the inner loop, which are different from prior works and are important in bilevel optimization. It fills the gap in the current literature.

**Weaknesses**

**1. The bound is loose.**

1.1)	The stability bounds derived in this paper seem to have worse dependence on the outer iteration number $K$ than [Bao et al. 2021]. It has exponential dependence in $K$ in the worst case in this paper while only polynomial dependence on $K$ in [Bao et al. 2021].

1.2)	No discussion of the tightness of the bound is provided in either theoretical or empirical form.

1.3)	It is contour-intuitive that when $f$ and $g$ are strongly convex, the stability has worse rate on $K$ (exponential) than when $f$ $g$ are non-convex (polynomial). I believe the bounds of SC-SC and C-C cases are not tight.

**2. Incorrectness**

2.1) Summary of the most relevant work [Bao et al. 2021] is incorrect.

a) My understanding for paper [Bao et al. 2021] is that it assumes the compositional function $f(x, \hat{y}(x, D_{m2}); \xi)$ w.r.t. $x$ is Lipschitz continuous, which also has a dependence in number of iterations $T$ in the inner loop. Therefore, the summary in Table 1 for UD[Bao et al. 2021] which does not depend on $T$ is not accurate. In addition, [Bao et al. 2021] provides the results of convex case in the appendix.

b) What is $\nabla f$? The comparison of UD[Bao et al. 2021] and TSGD in appendix A is not correct as the gradient of $x$ in the outer loop is not the same based on my understanding. Because it requires second-order information for UD[Bao et al. 2021].

2.1) Outer update/reference of SSGD and TSGD is incorrect.

a) In Algorithms 1 & 2, the update for outer parameter $x$ does not take gradient w.r.t. function $\hat{y}$? Could you add references to the two algorithms analyzed in this paper, for single timescale and two-time scale, separately?

b) If the update for outer parameter $x$ does not take gradient w.r.t. function $\hat{y}$, it is not the same as the algorithm for some two-time scale methods referenced in this paper such as [Ghadimi & Wang, 2018], [Ji et al. 2021] as they require the second-order information of the loss during the update of $x$.

**3. Lack of clarity**

a) How does the results in this paper compare with existing stability analysis on specific bilevel problems such as minmax problems and meta-learning?

b) Although different algorithm stability concepts are defined, only uniform-stable in expectation is used for the main theorems in Section 4. Why do you need to define uniform stability with probability $1-\delta$ in Definition 4, (a)?

c) Why do you need Theorem 1 (c)? Theorem 1 (c) is not used in Section 4.

d) The proofs are not clear. Some proofs of the TSGD case are omitted due to claimed similarity of the SSGD case (see pages 20-21). However, I suggest that you include proof for TSGD case and omit the SSGD case.

**3.2 Minor comments**

a) In Theorem 1 (c), what is $e$?

b) In Theorem 1 (c), A is uniform stable with probability at least $1 - \delta$?

c) Throughout the paper, why do you use $\nabla_y g$ for the update of $y$  and $\nabla f$ for the update of $x$ but not $\nabla_x f$ ?

d)	The notation $G$ is used for both inner population risk and update function, please use different notations to avoid confusion.


**Summary Of The Paper:**

This paper studies the generalization of bilevel optimization algorithms in the framework of algorithm stability. It focuses on two algorithms, namely, single-time scale (SSGD) and two-time scale (TSGD) with first-order stochastic gradient updates.

Different from prior work, the algorithms considered do not require reinitialization in the inner loop. Experiments on data reweighting and meta-learning are provided to show how the generalization performance changes w.r.t. the number of iterations.


**Summary Of The Review:**

This paper considers generalization in terms of algorithm stability for two bilevel algorithms SSGD, and TSGD, without reinitialization in the inner loop, which is different from prior works.

However, the TSGD algorithm described in the paper is different from what is referenced. Since the papers referenced all require second-order information in the update of outer parameters. But this paper does not require second-order information.

Also the summary in Table 1 is not reasonable for prior work [Bao et al. 2021], since it should also depends on $T$, the number of inner iterations.

For the above reasons, I recommend rejection.

---

> ### Author Response · Authors · 2022-11-18
> **Response to the Reviewer GoE2**
>
> We thank this reviewer for the thoughtful comments. Below are our responses to the raised concerns.
>
> - For stability bounds comparison, please see our response to common concern 1.
>
>    Our paper focuses on the stability bounds of bilevel optimization. We agree that it would be ideal to also determine the tight bounds. However, given the fact that estimating the upper bounds is already a challenging problem, we feel that finding the lower bounds would need some new techniques and thus will be a nice problem for future work.
>
>    We have recently improved our results for the strongly-convex-strongly-convex and convex-convex cases, which provide a novel perspective to estimate the stability bounds for the two-timescale algorithms. Please refer to Theorem 3 in the revised paper for details.
>
>    Compared to the single-level problem in existing work [2], bilevel optimization is much more challenging due to the facts that it involves two functions and the algorithm TSGD has a two-layer loop, while the problem in [2] has only one function and its simple SGD algorithm has only one layer in the loop. Because of such difference, the inner loop in our TSGD will accumulate gradients from multiple SGD steps, which could make some nice properties such as convexity or strong convexity extremely difficulty to be exploited. Following the framework of algorithmic stability analysis in [2], we investigate the bilevel problem by employing the expansivity of update rules (please see details in Lemma 13). In the single level problem, e.g. [2], gradient update is non-expansive in the convex case and contractive in the strongly-convex case. In TSGD, the inner function has an accumulation of multiple SGD steps, which may have a negative impact on these properties. Actually this phenomenon has also occurred in recent work [4] for minmax problems. This seems to be an interesting problem for future research to further examine this phenomenon.
>
> - $\nabla f = \nabla_x f(x_k, y_k^T, D_{m_2})$. We have revised the algorithms to fix the typos. It seems that [1] does not require second-order information for UD. If they do, there would be analysis on hypergradients, which involves second order matrices.
>
> - The reviewer may misunderstand our descriptions that we do know that $y(x)$ should be an argument depending on $x$, while the proof of Theorem 2 in [1] may lead to misconceptions. Please see details in our comments on common concern 1.
>
> - For minmax problems, the stability bounds analysis mainly focus on the SGDA and AGDA algorithms [3,4], which are different from ours. For meta-learning problems, the stability analysis is performed for some algorithms different from ours. Additionally, it is notable that due to the additional stochastic function in the constraint in the bilevel optimization, all the previous techniques and results cannot be applied to our problem. Minmax optimization involves only one objective function and a single level in algorithms for typical minmax optimization problems, while in the bilevel optimization algorithms there is an inner level and an outer level, which is considerably more challenging. Hence, it may not be suitable for us to compare these results.
>
> - We introduce uniform stability with high probability form because Theorem (1.c) establishes a general connection between generalization error and algorithmic stability for any randomized algorithms, not specified for our SSGD and TSGD.
>
> - Thanks for the suggestions. We add additional proofs in appendix.
>
> - $e$ is the base of the natural logarithms, approximately equal to 2.71828, and we add its meaning in the revised paper.
>
> - $G_s$ and $G_T$ are used for update rules.
>
>
> [1] Bao, Fan, et al. "Stability and Generalization of Bilevel Programming in Hyperparameter Optimization." Advances in Neural Information Processing Systems 34 (2021): 4529-4541.
>
> [2] Hardt, Moritz, Ben Recht, and Yoram Singer. "Train faster, generalize better: Stability of stochastic gradient descent." International conference on machine learning. PMLR, 2016.
>
> [3] Lei, Yunwen, et al. "Stability and generalization of stochastic gradient methods for minimax problems." International Conference on Machine Learning. PMLR, 2021.
>
> [4] Farnia, Farzan, and Asuman Ozdaglar. "Train simultaneously, generalize better: Stability of gradient-based minimax learners." International Conference on Machine Learning. PMLR, 2021.

---

### Official Review · Reviewer_rxeE · 2022-10-27

**Confidence:** 4
**Correctness:** 3
**Technical Novelty And Significance:** 3
**Empirical Novelty And Significance:** 2
**Recommendation:** 5

**Clarity, Quality, Novelty And Reproducibility:**

Please see the section 'Clarity, Quality, Novelty And Reproducibility'.


**Strength And Weaknesses:**

## Strength

1. Overall, this paper is very well written.
2. This paper is sound since the claims are supported by formal theoretical results. Besides, experimental results are also provided to corroborate the theory.
3. The proofs seem reasonable and right although I have not checked them line-by-line.
4. It clearly discusses the differences with related work, especially the previous work[1*].




## Weaknesses

1. The expression about "generalization error" is not suitable. In this paper, the authors define the bilevel generalization error as the difference between the population risk and empirical risk. It is strange because in statistical learning theory,
the generalization error usually refers to the population risk and the related expression in this paper can be confusing. I suggest the authors use the generalization gap, just as in previous work[1*].
2. While this paper refines the generalization analyses over previous work[1*], few additional insights can be offered to explain the practical phenomena and design effective learning algorithms in comparison with previous work[1*].
3. I do not agree with the expression about the limitation of the previous work[1*] (mainly in Remark 4). [1*] made an assumption that the update of $y$ in the inner level after the reinitialization will not be affected by the value specified for $x$, which may have a gap with the practical algorithms.
But, the main reason is to decouple the variables $x$ and $y$, and the proof is technically right.
4. As discussed in 3, how the authors deal with the case when updating the outer variable $x$ but the inner variable $y$ can be dependent on the $x$ and the hyper-gradient is used. Please give more explanations and discussions.



[1*]Fan Bao, Guoqiang Wu, Chongxuan Li, Jun Zhu, and Bo Zhang. Stability and generalization of bilevel programming in hyperparameter optimization. NeurIPS 2021

**Summary Of The Paper:**

This paper studies the generalization of bilevel optimization problems which are widely used in machine learning, e.g., meta-learning, hyper-parameter optimization, and
reinforcement learning. Specifically, the authors conduct a thorough analysis of the generalization of first-order (gradient-based) methods for the bilevel optimization problem.
Technically, in comparison with the previous work, the (advanced) algorithmic stability tool is used to give a high probability generalization bound which improves the previous best one $\sqrt(n)$
to $\log(n)$, where $n$ is the sample size. Besides, some particular settings (e.g., strongly-convex-strongly-convex (SC-SC)) are also involved. Finally, experimental results are provided to support the
theoretical results.


**Summary Of The Review:**

In summary, this paper conduct refined generalization analyses for the bilevel optimization problems over previous work via advanced stability techniques. But, some necessary discussions and revisions should be made.

---

> ### Author Response · Authors · 2022-11-18
> **Response to the Reviewer rxeE**
>
> We thank the reviewer for the thoughtful comments. Below are our responses to the raised concerns.
>
> - We thank the reviewer for this helpful suggestion. We have revised our paper accordingly to better follow up on the previous work [1].
>
> - Please see our response to common concern 1 to address the concern.
>
> [1] Bao, Fan, et al. "Stability and Generalization of Bilevel Programming in Hyperparameter Optimization." Advances in Neural Information Processing Systems 34 (2021): 4529-4541.v

---

### Author Response · Authors · 2022-11-18
**Common concerns from reviewers**

We are very grateful to all the reviewers for their insightful and constructive comments.  We have revised our paper accordingly.
Below are our responses to some common concerns.

- Common concern 1 from the reviewers [rxeE, Wyf6, Goe2]: misunderstanding that UD treats $y(x)$ as an argument independent $x$.

    We thank all the reviewers for pointing out this important issue. It seems that the reviewers misunderstood our description about the previous work [1]. We do know that $y(x)$ (which corresponds to $\theta(\lambda)$ in [1]) should be an argument depending on $x$ (or $\lambda$ in [1]), which is consistent with all the reviewers' belief. However, the proofs in [1] make us suspect that $\theta(\lambda)$ is more or less treated as an argument independent of $\lambda$ in that paper, even though in the description $\theta(\lambda)$ is said to be dependent on $\lambda$. Below are some evidence.
    In the proof of Theorem 2 (Appendix A.2 of [1], page 14),  the following equations are given,
    $$
    \begin{aligned}
    &\ell\left(\mathbf{A}\left(S^{t r}, S^{v a l}\right), z\right)=\ell\left(\lambda_t, \hat{\theta}\left(\lambda_t, S^{t r}\right), z\right)=f\left(\lambda_t, \hat{\theta}\right) \\
    &\ell\left(\mathbf{A}\left(S^{t r}, S^{\prime v a l}\right), z\right)=\ell\left(\lambda_t^{\prime}, \hat{\theta}\left(\lambda_t^{\prime}, S^{t r}\right), z\right)=f\left(\lambda_t^{\prime}, \hat{\theta}\right).
    \end{aligned}
    $$
    This implies that $\hat{\theta}\left(\lambda_t, S^{t r}\right) = \hat{\theta}$ and $\hat{\theta}\left(\lambda_t^{\prime}, S^{t r}\right)= \hat{\theta}$, which seems to suggest that $\hat{\theta}(\lambda)$ is independent of $\lambda$ and thus  conflicts with their claim that $\theta(\lambda)$ is dependent on $\lambda$.

    One possible explanation is that [1] may abuse the notations slightly. But that possibility is not supported by their following proof, which is quite confusing.

    To understand the issue better, let $\delta_t=\left\|\lambda_t-\lambda_t^{\prime}\right\|$. Suppose that $0 \leq t_0 \leq t$.  They have the following inequality in the proof.
    $$
    \begin{aligned}
    \mathbf{E}\left[\left|f\left(\lambda_t, \hat{\theta}\right)-f\left(\lambda_t^{\prime}, \hat{\theta}\right)\right|\right]=& \mathbf{E}\left[\left|f\left(\lambda_t, \hat{\theta}\right)-f\left(\lambda_t^{\prime}, \hat{\theta}\right)\right| \cdot 1_{\delta_{t_0}=0}\right] +\mathbf{E}\left[\left|f\left(\lambda_t, \hat{\theta}\right)-f\left(\lambda_t^{\prime}, \hat{\theta}\right)\right| \cdot 1_{\delta_{t_0}>0}\right] \\
    \leq & L \mathbf{E}\left[\delta_t \cdot 1_{\delta_{t_0}=0}\right]+P\left(\delta_{t_0}>0\right) s(\ell) .
    \end{aligned}
    $$

    The left hand side is to measure the expected difference of function $f$ with arguments $(\lambda_t, \hat{\theta})$. The first equation decomposes the left hand side according to the two possible cases of  $\delta_{t_0}$ (i.e., $\delta_{t_0}=0$ or $\delta_{t_0} > 0$). The first term of the inequality is derived from the Lipschitz continuous property of $f$. However, to use the Lipschitz continuity of the multivariate function $f$ with respect to $\hat{\theta}$, i.e.,
    $|f(\lambda_t,\hat{\theta}) - f(\lambda_t^{\prime},\hat{\theta}) | \leq L\| \lambda_t -\lambda_t^{\prime}\|,$

    $\hat{\theta}$ needs to be the same varible (i.e., they have the same value all the time) in both $f(\lambda_t,\hat{\theta})$ and $f(\lambda_t^{\prime},\hat{\theta})$.

    However, from the UD algorithm in [1] it is clear that $\hat{\theta}$ will not always have the same value in $f(\lambda_t,\hat{\theta})$ and $f(\lambda_t^{\prime},\hat{\theta})$ when $\lambda$ changes.

    Specifically, when $t=t_0, \delta_{t_0} =0, \lambda_{t_0} = \lambda_{t_0}^{\prime}$, we have

    $\hat{\theta}^{{t_0 }+1}_K (\lambda _{t_0}, S^{tr}) = {\hat{\theta}^{\prime}}^{{t_0}+1}_K (\lambda^{\prime} _{t_0}, S^{tr})$

    Since $S^{val}$ and ${S^{\prime}}^{val}$ are assumed to differ by at most one point,  without loss of generality, we suppose that SGD selects the different point at timestep $t_s$. Then, we have

    $t=t_s$

    $\lambda_{t_s} \neq \lambda_{t_s}^{\prime} $

    $\delta_{t_s} \neq 0$

     $\hat{\theta}^{{t_s }+1}_K (\lambda _{t_s}, S^{tr}) \neq {\hat{\theta}^{\prime}}^{{t_s}+1}_K (\lambda^{\prime} _{t_s}, S^{tr})$

    This means that  $\hat{\theta}^{{t_s }+1}_K (\lambda, S^{tr}) \neq {\hat{\theta}^{\prime}}^{{t_s}+1}_K (\lambda^{\prime}, S^{tr})$  for all $t\ge t_s$, as the update of $\hat{\theta}$ and $\lambda$ use the value of $\hat{\theta}$ in the previous iteration. Thus, we cannot use the Lipschitz property to derive the first term in the aforementioned inequality.  That is why we think there may exist some gap in the analysis of UD algorithms in [1].
[1] Bao, Fan, et al. "Stability and Generalization of Bilevel Programming in Hyperparameter Optimization." Advances in Neural Information Processing Systems 34 (2021): 4529-4541.

---

> ### Comment · Reviewer_Wyf6 · 2022-11-20
> **Thanks for the reply**
>
> Indeed, the proof still treats $\theta(\lambda)$ as a function of $\lambda$.
>
> Here $f$ is a mapping that takes a vector $\lambda$ and a function $\theta(\cdot)$ as input. Note that $\theta$ in $f(\lambda, \theta)$ denotes the function $\theta(\cdot)$ instead of a vector.

---

> > ### Author Response · Authors · 2022-12-07
> > **Thanks for the reply**
> >
> > Thanks for your response, and there is one more question that bothers us.
> > - If here $\theta$ would be treated as the function of $\lambda$ in the $f(\lambda, \theta)$, do we need to consider the properties of $\theta$ function for $\lambda$ (how the properties will have an impact on the $f$ with respect to $\lambda$).

---

> > > ### Comment · Reviewer_Wyf6 · 2022-12-07
> > > **Response**
> > >
> > > Yes. The property of $\theta(\cdot)$ is derived according to Assumption 1-4 in the proof of Theorem 3 in [1]. It provides properties such as Lipschitz continuity.

---

> > > > ### Author Response · Authors · 2022-12-07
> > > > **Thanks for the response**
> > > >
> > > > Thanks for your quick reply.
> > > >
> > > > We would concern whether the properties of the $\theta$ function will have an impact on the $f$ since here the paper considers the Lipschitz continuity and smoothness of $f$ with respect to $\lambda$, which seems to indicate the $\theta$ function is vacuous.

---

> > > > > ### Comment · Reviewer_Wyf6 · 2022-12-12
> > > > > **Response**
> > > > >
> > > > > I agree this paper considers the Lipschitz continuity $L$ and smoothness $\gamma$ of $f$ with respect to $\lambda$. However, this does not indicate the $\theta$ function is vacuous.
> > > > >
> > > > > Indeed, in Theorem 3, the authors have analyzed what $L$ and $\gamma$ are for the unrolled differentiation algorithm.
> > > > > $L$ and $\gamma$ are related to Lipschitz properties of the $\theta$ function, which are provided in Lemma 3 and Lemma 4. In the proof of Theorem 3, the authors derive $L$ and $\gamma$ according to the Lipschitz properties of the $\theta$ function.

---

### Author Response · Authors · 2022-11-18
**Common concerns (2)**

Due to space limitations, we provide more responses here.

- Common concern 2 from the reviewers [Wyf6, GoE2]: Why a stronger assumption leads to a looser bound?

    We have recently improved our results for the strongly-convex-strongly-convex and convex-convex cases, which provide a novel perspective to estimate the stability bounds for the two-timescale algorithms. Please refer to Theorem 3 in the revised paper for details.

    Compared to the single-level problem in existing work [2], bilevel optimization is much more challenging due to the facts that it involves two functions and the algorithm TSGD has a two-layer loop, while the problem in [2] has only one function and its simple SGD algorithm has only one layer in the loop. Because of such difference, the inner loop in our TSGD will accumulate gradients from multiple SGD steps, which could make some nice properties such as convexity or strong convexity extremely difficulty to be exploited. Following the framework of algorithmic stability analysis in [2], we investigate the bilevel problem by employing the expansivity of update rules (please see details in Lemma 13). In the single level problem, e.g. [2], gradient update is non-expansive in the convex case and contractive in the strongly-convex case. In TSGD, the inner function has an accumulation of multiple SGD steps, which may have a negative impact on these properties. Actually this phenomenon has also occurred in recent work [3] for minmax problems. This seems to be an interesting problem for future research to further examine this phenomenon.

- We have fixed all minor issues and typos in the revised version. Thanks for pointing out those issues.

- We reorganized section 1.

- We added an additional experiment on MNIST dataset according to suggestions from the Reviewer jxJW  in Appendix D.1

[1] Bao, Fan, et al. "Stability and Generalization of Bilevel Programming in Hyperparameter Optimization." Advances in Neural Information Processing Systems 34 (2021): 4529-4541.

[2] Hardt, Moritz, Ben Recht, and Yoram Singer. "Train faster, generalize better: Stability of stochastic gradient descent." International conference on machine learning. PMLR, 2016.

[3] Farnia, Farzan, and Asuman Ozdaglar. "Train simultaneously, generalize better: Stability of gradient-based minimax learners." International Conference on Machine Learning. PMLR, 2021.

---

### Decision · Program_Chairs · 2023-01-20

**Decision:**

Reject

**Justification For Why Not Higher Score:**


There are some improper or incorrect claims on the prior work, upon which this work is built.
The technique in this work is lacks novelty, since its technique mainly follows papers mentioned in the reviews.

**Justification For Why Not Lower Score:**

A well written paper.

**Metareview: Summary, Strengths And Weaknesses:**


This paper studies the generalization performance of bilevel optimization problems.
The paper considers two algorithms SSGD, and TSGD without reinitialization in the inner loop, which are different from prior works and are important in bilevel optimization. It is easy to follow, yet the novelty appeared not significant enough.